# EDMolGPT: A Decoder-Only Framework for 3D Drug Design via Electron Density

## Abstract

Electron density-guided drug design is a promising structure-based drug discovery (SBDD) frontier, crucial for delineating dynamic molecular features and intermolecular interactions. Existing methods leveraging electron density for *de novo* molecule generation employ a two-stage process: generating hypothetical binder electron densities within a pocket, then interpreting them into molecules. While mitigating bias from binders pre-existing in the pocket, these approaches' two-stage nature can lead to error accumulation. Furthermore, these methods are limited by rigid pocket assumptions, which may compromise the diversity of the generated electron density. These limitations often result in drug-like molecules lacking favorable three-dimensional (3D) conformations or conversely, 3D conformations without assured drug-likeness. We introduce EDMolGPT, a novel decoder-only framework that directly synthesizes molecules from the low-resolution electron density point cloud derived from an existing binder. By leveraging this existing binder's low-resolution electron density and avoiding explicit pocket structures, our strategy effectively mitigates bias, circumvents two-stage error, and negates rigid pocket limitations. EDMolGPT's autoregressive decoder-only architecture, guided by robust low-resolution electron density, efficiently generates binding molecules with high drug-likeness and favorable 3D conformations. Rigorous validation across 101 biological targets underscores its potential to accelerate novel therapeutic agent discovery.

## 1 Introduction

Generative AI models applied in structure-based drug design (SBDD) have revolutionized the field for their ability to generate ligands spatially compatible with a binding pocket's 3D architecture. Despite this success, most current AI molecule generation models for SBDD overlook the dynamic nature of binding sites by assuming a static pocket representation (Feng et al., 2024; Qu et al., 2024). As illustrated in Fig. 1, modeling the pocket as a rigid structure fails to capture the

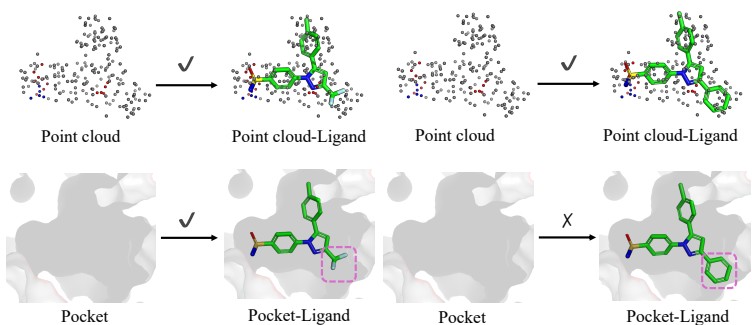

(a) Generation under rigid structure   (b) Generation under flexible structure

Figure 1: We present the generation pipeline based on the pocket (pdb_id=3L1N) and electron density. (a) present that this molecule can be generated via both the electron density while the molecule presented in (b) can only be generated via the electron density. The region highlighted in purple indicates the flexible region (valid via the wet experiments), and the electron density can induce the generation of larger fragments at this location while maintaining the activity.

intrinsic flexibility of proteins and their conformational changes upon ligand binding (Lu et al., 2024). Such oversimplified representations create a mismatch between the modeled pocket and

its biologically relevant states, posing significant challenges for accurate molecule generation and potentially reducing the success rate of identifying truly active compounds.

To overcome these rigid pocket limitations, researchers are exploring conditioning signals that can more naturally describe binding site dynamics and guide molecule generation. Among various candidate representations, electron density (ED) emerges as a highly informative descriptor, inherently encoding the binding site's spatial distribution, its physicochemical environment, critical intermolecular interactions, and crucial pocket flexibility. As shown in Fig. 1 leveraging electron density as a conditional input allows models to capture subtle steric, electrostatic, and non-covalent interaction features across multiple conformational states, enabling the generation of molecules that are not only geometrically compatible but also chemically and biologically meaningful (Ding et al., 2022b; Ma et al., 2023). Differently, the rigid pocket cannot generate this case shown in Fig. 1 right since the molecule collides with the pocket of this conformation.

Nevertheless, existing AI molecule generation models have not fully exploited the potential of electron density. Wang's Model (Wang et al., 2022), the first AI generative model using electron density as molecule representation, established a two-stage paradigm: first generating electron densities for potential binders within a pocket, then interpreting these into molecular structures. While this approach helps mitigate bias from pre-existing binders, its two-stage nature can lead to error accumulation of ED intensity, as shown in Fig. 2. Such limitations cause the model not able to simultaneously fulfill drug-likeness and favorable 3D conformation of generated molecules. As a result, Wang's Model used a fragment library for molecule assembly, which limits the diversity of output. The following work ED2Mol (Li et al., 2025) treats electron density as an auxiliary constraint and assembles molecules also by selecting fragments from a predefined library, facing the same challenge of generating novel and diverse scaffolds.

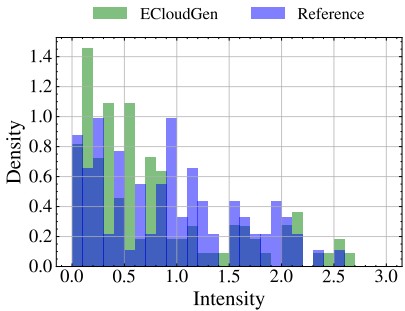

Figure 2: The distribution of ED intensity. Green and blue denote the ED estimated via ECloudGen and extracted from the Reference ligand, respectively. There is a significant difference between the predicted ED and the actual ED.

Other research, such as ECloudGen (Zhang et al., 2024), conditions a sequence-generation model on electron density. While successfully enabling *de novo* molecule design without relying on a fragment library, it neglects explicit 3D structural reasoning, which may compromise the geometric plausibility of the output. These observations motivate the need for a novel electron-density–guided generative framework that directly operates in 3D space and better captures the flexible nature of protein pockets, ultimately improving the accuracy and success rate of molecule discovery.

Ma et al. (Ma et al., 2023) demonstrated that low-resolution electron density of a pre-existing binder can mitigate bias derived from this reference binder in structure-based virtual screening. Leveraging this insight, we construct low-resolution electron density point clouds directly from a known binding molecule and use this electron density as conditional input for molecule generation. Specifically, given a binding molecule, we compute its electron diffraction pattern using the Fast Fourier Transform (FFT) and apply a high-resolution cutoff to control the level of structural detail. The inverse transform then yields a smooth electron density map, from which we uniformly sample a fixed number of grid points to obtain a low-resolution ED point cloud representation, which offers several key advantages: it effectively preserves the spatial distribution of the binding environment without introducing bias from the structural details of the pre-existing binder; it inherently avoids the two-stage generation paradigm of previous models, thereby preventing associated error accumulation; and it circumvents the rigid assumption of pocket structure, as shown in Fig. 1 (b). To further enhance chemical relevance, we label each point in the cloud with pharmacophore features such as hydrogen bond donor, hydrogen bond acceptor, donor–acceptor, or other, thereby providing a rich physicochemical context that guides the generative model toward producing biological molecules.

Based on low-resolution ED as rationally guided information, we propose EDMolGPT, a decoder-only autoregressive framework for 3D drug design conditioned on the extracted point cloud. Given the importance of input order in GPT-style models, we reorder the point cloud according to spatial coordinates to provide a consistent and meaningful sequence. To represent the generated molecules, we adopt FSMILES (Feng et al., 2024), which has been demonstrated to effectively capture rational molecular conformations. Unlike most existing ligand generation frameworks, which predominantly

rely on encoder–decoder architectures or diffusion models, our approach is the first to apply a decoder architecture to this task. This design combines simplicity, flexibility, and high generation efficiency, while fully leveraging model parameters to produce accurate 3D molecular structures. To validate the effectiveness of EDMolGPT, we perform experiments on the DUD-E dataset and evaluate performance across multiple metrics. The results show that our framework not only generates molecules with conformations compatible with the given binding pocket but also produces compounds with bioactivity. Our contribution can be summarized as follows:

- **Novel Electron Density-Guided Framework:** We introduce EDMolGPT, to our knowledge, the first decoder-only autoregressive model for 3D drug design that is conditioned on low-resolution electron density representing the binding environment. This approach addresses the limitations of rigid pocket representations and two-stage generation paradigms prone to error accumulation.

- **Mitigation of Bias and Rigidity:** By utilizing low-resolution electron density derived from pre-existing binders and avoiding explicitly using pocket structures, EDMolGPT mitigates bias associated with reference binder details and circumvents the rigid pocket assumption, allowing for the generation of molecules compatible with the dynamic nature of protein binding sites.

- **Validated Performance:** Through extensive experiments on up to 101 targets from DUD-E dataset, we show that EDMolGPT consistently generates molecules with both favorable 3D conformations compatible with the target binding pocket and demonstrated bioactivity, validating its potential for *de novo* drug discovery.

## 2 RELATED WORK

**Structure-based drug design** SBDD generates ligands by exploiting the 3D structure of a target receptor. Classical SBDD workflows, such as molecular docking (Morris et al., 2009), scoring functions (Breda et al., 2008), and molecular dynamics (MD) simulations (Hollingsworth & Dror, 2018), are computationally expensive, particularly for large-scale virtual screening. To address these limitations, recent advances integrate AI-based generative modeling, with progress in both autoregressive (Gao et al., 2022) and diffusion-based approaches (Xu et al., 2022). Among autoregressive methods, Pocket2Mol (Peng et al., 2022) introduced an E(3)-equivariant generative framework that directly samples valid molecules from pocket geometry, improving affinity and diversity. Lingo3DMol (Feng et al., 2024) further incorporated fragment-based SMILES with 3D geometric features to enable language-model-driven molecule generation. In diffusion-based methods, TargetDiff (Guan et al., 2023a) conditions on protein pocket information to generate ligands with high binding affinity, while MolCRAFT (Qu et al., 2024) performs noise-controlled sampling for stable conformations and superior docking scores. Different from them, our EDMolGPT generates full 3D ligand conformations conditioned on the point clouds extracted from low-resolution electron density. This enables the generation of novel valid molecules with accurate structural geometry.

**Electron density-guided molecule generation** Recent advances incorporate electron density (ED) into AI-driven molecule generation, yet existing methods trade off scaffold novelty, 3D conformation fidelity, and the balance between drug-likeness and binding constraints. Wang et al. (Wang et al., 2022) introduced the first ED-guided generative model with a two-stage pipeline: generating ligand electron densities in protein pockets, then translating them into molecules via fragment-based assembly. This reduces dependence on known ligands but may propagate errors and limit scaffold diversity. ED2Mol (Li et al., 2025) uses ED as an auxiliary constraint for fragment-based assembly guided by density and drug-likeness, improving plausibility but still restricting scaffold exploration. ECloudGen (Zhang et al., 2024) conditions a sequence-generation model on ED for *de novo* design without fragments, yet lacks explicit 3D reasoning, risking binding conformation quality. In contrast, our method directly leverages low-resolution electron density as a continuous 3D field to guide atomic placement in an end-to-end manner, avoiding fragment libraries and simultaneously ensuring drug-likeness, geometric fidelity, and binding compatibility for novel scaffold generation.

## 3 METHOD

In Sec. 3.1, we formulate the problem of electron density-based drug design. Building upon this, Sec. 3.2 describes the extraction of point clouds from electron density. In Sec. 3.3, we present the

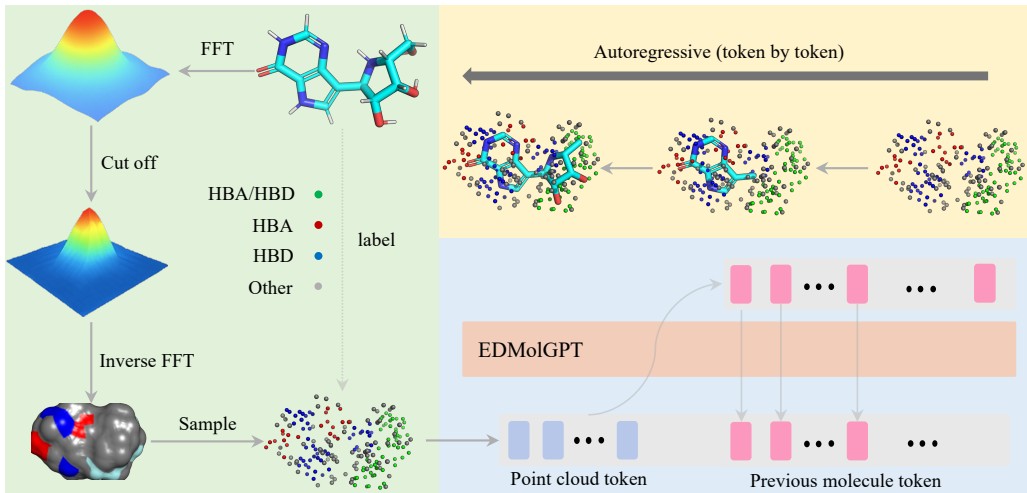

Figure 3: The overall pipeline of our method. The components shown with a green background correspond to the generation of 3D point clouds from the input ligand. The blue-highlighted components represent the molecule generation process, where each molecular token is predicted sequentially based on the point cloud tokens and the previously generated molecular tokens. Finally, the steps highlighted in yellow illustrate how the predicted molecular structure progressively occupies and fills the sampled point clouds, providing an interpretable view of the generation process.

representation of molecular structures, including FSMILES and relative distances. Finally, Sec. 3.4 details the overall EDMolGPT architecture and the procedures for training and inference, specifically how to generate a molecule conditioned on a given point cloud.

## 3.1 PROBLEM FORMULATION

The goal of drug design is to generate a molecule $\mathcal{M} = \{(a_m^i, \boldsymbol{v}_m^i)\}_{i=1}^{N_m}$, which consists of $N_m$ atoms, where $a_m^i \in \mathbb{R}^1$ denotes the atom type and $\boldsymbol{v}_m^i \in \mathbb{R}^3$ represents its position in 3D space, with three components corresponding to the $x$, $y$, and $z$ coordinates. Traditional SBDD conditions molecule generation directly on the full receptor structure $\mathcal{R}$ to produce $\mathcal{M}$. Different from traditional SBDD, our method conditions on point clouds extracted from the binding molecule itself. Specifically, we construct a compact point cloud representation $\mathcal{P}_m = \{(c_p^i, \boldsymbol{v}_p^i)\}_{i=1}^{N_p}$ from $\mathcal{M}$, where $c_p^i$, $\boldsymbol{v}_p^i$, and $N_p$ denote the point types, coordinates, and number of points, respectively. This geometric representation provides a rich yet compact conditioning signal, enabling the model to capture the binding context sufficiently.

## 3.2 GENERATING POINT CLOUD

Given a binding molecule, we first represent its structure in reciprocal space by computing the electron diffraction pattern using the Fast Fourier Transform (FFT). To control the spatial resolution, a high-frequency cutoff is applied based on the minimum interplanar spacing $d_{\min}$, such that only spatial frequencies corresponding to features larger than $d_{\min}$ are retained. The filtered diffraction data are then transformed back into real space to reconstruct a smooth electron density, from which 3D point clouds are sampled (Fig. 3). Formally, let $\{\boldsymbol{v}_m^i\}_{i=1}^{N_m}$ denote the atomic coordinates of molecule $\mathcal{M}$. The corresponding structure factors in reciprocal space are computed as

$$F(\boldsymbol{h}) = \sum_{i=1}^{N_m} f_i(\boldsymbol{h}) \, e^{2\pi i \boldsymbol{h} \cdot \boldsymbol{v}_m^i}, \tag{1}$$

where $\boldsymbol{h}$ is the reciprocal-lattice vector and $f_i(\boldsymbol{h})$ is the atomic scattering factor of atom $i$. Motivated by previous work showing that low-resolution electron density of pre-existing binders can mitigate bias in structure-based virtual screening (Ma et al., 2023), we apply a high-frequency cutoff and

retain only components with $|\boldsymbol{h}| \leq 1/d_{\min}$. The electron density is then obtained by truncating:

$$\rho(\boldsymbol{v}_m) = \frac{1}{V} \sum_{|\boldsymbol{h}| \leq 1/d_{\min}} F(\boldsymbol{h}) \, e^{-2\pi i \boldsymbol{h} \cdot \boldsymbol{v}_m}. \tag{2}$$

This truncation removes the high-frequency components while retaining the low-frequency terms, resulting in a smoothed, low-resolution electron density. We then randomly sample $N_p$ points from $\rho(\boldsymbol{v}_m)$ to generate a set of low-resolution ED point clouds $\{\boldsymbol{v}_p^i\}_{i=1}^{N_p}$. While these point clouds capture the overall molecular structure of the ligand, they contain limited pharmacophore information, posing challenges for the generation of bioactive molecules. To enrich the chemical features, for each point in the cloud, we compute its minimal distance to all atoms in the molecule $\mathcal{M}$ and assign a pharmacophore type based on the closest atom. Specifically, each point is assigned a type indicator $c_p^i$, whose value is selected from {hydrogen bond donor (HBD), hydrogen bond acceptor (HBA), hydrogen bond donor/acceptor (HBD/HBA), Other}. Finally, we obtain a set of labeled point clouds $\mathcal{P}_m = \{(c_p^i, \boldsymbol{v}_p^i)\}_{i=1}^{N_p}$. Since autoregressive models are sensitive to input ordering, we sort the points in $\mathcal{P}_m$ in ascending order of their $x$, $y$, and $z$ coordinates, thereby providing a consistent and spatially meaningful sequence for EDMolGPT.

## 3.3 INPUT FORMAT OF MOLECULE

Determining the ordering of the molecular structure $\mathcal{M}$ is crucial for autoregressive modeling. A straightforward approach is to use SMILES (Weininger, 1988) to represent the molecule together with its absolute spatial positions. While this representation is sufficient to describe molecular structures, its application in autoregressive generation often results in unrealistic or physically inconsistent conformations (Feng et al., 2024; Qu et al., 2024). To overcome this limitation, we adopt a modified Lingo3DMol representation (Feng et al., 2024) for $\mathcal{M}$, yielding $\widehat{\mathcal{M}} = \{(\widehat{a}_m^i, \widehat{\boldsymbol{v}}_m^i, \widehat{l}_m^i, \widehat{\theta}_m^i, \widehat{\phi}_m^i)\}$, where $\widehat{a}_m^i$, $\widehat{\boldsymbol{v}}_m^i$, $\widehat{l}_m^i$, $\widehat{\theta}_m^i$, and $\widehat{\phi}_m^i$ denote the Fragment SMILES (FSMILES) token, discretized 3D coordinates, bond length, bond angle, and dihedral angle, respectively. In the following section, we detail the procedure for converting a given molecule $\mathcal{M}$ into its representation $\widehat{\mathcal{M}}$.

**FSMILES** FSMILES (Feng et al., 2024) is a novel 2D molecular representation derived from SMILES, which decomposes molecules into fragments while retaining the standard SMILES syntax for each fragment. Compared with SMILES, FSMILES improves the learning of 2D molecular patterns by representing fragments and local structures with dedicated symbols and by prioritizing ring closures, which facilitates the generation of molecules with correct ring structures and bond angles. However, in the original FSMILES, edges connecting atoms within a ring were often cut, which could lead to overly fragmented molecular representations. Therefore, we improve FSMILES, avoiding splitting small fragments that link rings, thereby reducing excessive fragmentation and preserving more of the molecule's structural integrity. More details are in the Appendix Sec. B.1

**Discretized 3D coordinates** The coordinates of a molecule $\mathcal{M}$ are originally continuous in three-dimensional space. To make them compatible with autoregressive modeling, we discretize the spatial coordinates following the input format of Lingo3DMol. Specifically, we first compute the geometric center of $\mathcal{M}$, denoted as $\boldsymbol{\mu}_m \in \mathbb{R}^3$, and obtain the initial discretized coordinates:

$$\widetilde{\boldsymbol{v}}_m^i = \left\lfloor \frac{\boldsymbol{v}_m^i - \boldsymbol{\mu}_m}{\sigma} \right\rfloor, \tag{3}$$

where $\sigma$ and $\lfloor \cdot \rfloor$ denote a scaling factor and rounding operation, respectively. Since the spatial extent of most drug-like molecules lies within 5–30 Å, we set $\sigma = 0.1$, mapping the coordinates into a bounded integer grid of moderate resolution (within $[-150, 150]$ along each axis). To facilitate autoregressive prediction, we shift all discretized coordinates by a constant offset so they become positive integers. The final coordinates, denoted as $\widehat{\boldsymbol{v}}_m^i$, preserve geometric detail while keeping the vocabulary size manageable, thereby improving tractability and training stability. Point clouds are also transformed into this shifted space, denoted as $\{\widehat{\boldsymbol{v}}_p^i\}$. More details are in Appendix Sec. B.2.

**Relative Distance** Although the discretized coordinates $\boldsymbol{v}_m^i$ capture the absolute spatial positions of atoms, we further incorporate relative geometric information to explicitly model local structural

dependencies, which is beneficial for autoregressive inference. Specifically, for each atom $\boldsymbol{v}_m^i$, we consider its three preceding atoms $\boldsymbol{v}_m^{i-1}$, $\boldsymbol{v}_m^{i-2}$, and $\boldsymbol{v}_m^{i-3}$, and compute the bond length $l_m^i$, bond angle $\theta_m^i$, and dihedral angle $\phi_m^i$ as follows:

$$l_m^i = \left\| \boldsymbol{v}_m^i - \boldsymbol{v}_m^{i-1} \right\|_2,$$

$$\theta_m^i = \arccos\left( \frac{\left(\boldsymbol{v}_m^{i-1} - \boldsymbol{v}_m^{i-2}\right) \cdot \left(\boldsymbol{v}_m^i - \boldsymbol{v}_m^{i-1}\right)}{\left\| \boldsymbol{v}_m^{i-1} - \boldsymbol{v}_m^{i-2} \right\|_2 \left\| \boldsymbol{v}_m^i - \boldsymbol{v}_m^{i-1} \right\|_2} \right), \tag{4}$$

$$\phi_m^i = \arctan\Big( \left((\boldsymbol{b}_1 \times \boldsymbol{b}_2) \times (\boldsymbol{b}_2 \times \boldsymbol{b}_3)\right) \cdot \frac{\boldsymbol{b}_2}{\|\boldsymbol{b}_2\|_2}, (\boldsymbol{b}_1 \times \boldsymbol{b}_2) \cdot (\boldsymbol{b}_2 \times \boldsymbol{b}_3) \Big),$$

where $\boldsymbol{b}_1 = \boldsymbol{v}_m^{i-2} - \boldsymbol{v}_m^{i-3}$, $\boldsymbol{b}_2 = \boldsymbol{v}_m^{i-1} - \boldsymbol{v}_m^{i-2}$, and $\boldsymbol{b}_3 = \boldsymbol{v}_m^i - \boldsymbol{v}_m^{i-1}$. We similarly convert $l_m^i$, $\theta_m^i$, and $\phi_m^i$ into discrete representations for autoregressive modeling:

$$\widehat{l}_m^i = \left\lfloor \frac{l_m^i}{\sigma} \right\rfloor, \quad \widehat{\theta}_m^i = \left\lfloor \frac{\theta_m^i}{10} \right\rfloor, \quad \widehat{\phi}_m^i = \left\lfloor \frac{\phi_m^i}{10} \right\rfloor \tag{5}$$

As shown in Eq. 5, we discretize bond lengths using the same rule as coordinates. For bond angles and dihedral angles, we apply a coarser discretization, dividing the 180-degree range into 10-degree intervals. This design ensures that relative geometric information contributes effectively while preserving the learnability of the task. More details are in the Appendix Sec. B.3

## 3.4 EDMOLGPT

**Training** The overall architecture of EDMolGPT follows GPT-2 (Radford et al., 2019), a decoder-only framework. In our method, we adopt the default Transformer positional embeddings as used in GPT, which are learnable embeddings. During training, we concatenate the point cloud and molecule sequences and feed them into EDMolGPT to predict the molecule token-by-token. Formally, after acquiring the discretized point cloud $\widehat{\mathcal{P}}_m$ and the corresponding molecule $\widehat{\mathcal{M}}$, the input features for the point cloud and molecule are defined as:

$$\begin{aligned} \boldsymbol{h}_p^i &= g_x(\widehat{v}_{p,x}^i) + g_y(\widehat{v}_{p,y}^i) + g_z(\widehat{v}_{p,z}^i) + g_c(c_p^i), \\ \boldsymbol{h}_m^i &= g_x(\widehat{v}_{m,x}^i) + g_y(\widehat{v}_{m,y}^i) + g_z(\widehat{v}_{m,z}^i) + g_a(\widehat{a}_m^i), \end{aligned} \tag{6}$$

where $g_x$, $g_y$, $g_z$, $g_c$, and $g_a$ denote the embedding functions for X-, Y-, and Z-coordinates, point cloud type, and FSMILES token, respectively. Note that the coordinate embedding functions are shared between the point cloud and molecule, as they reside in the same spatial space. Since all items are converted into discrete space, we can employ linear classifiers to predict $\widehat{a}_m^t$, $\widehat{v}_m^t$, $\widehat{l}_m^t$, $\widehat{\theta}_m^t$, $\widehat{\phi}_m^t$, and optimize the model using the cross-entropy loss. The overall training objective is:

$$\mathcal{L}_{\text{EDMolGPT}} = -\frac{1}{N_m} \sum_{t=1}^{N_m} \log p\Big( (\widehat{a}_m^t, \widehat{v}_m^t, \widehat{l}_m^t, \widehat{\theta}_m^t, \widehat{\phi}_m^t) \;\Big|\; \underbrace{\boldsymbol{h}_p^1, \dots, \boldsymbol{h}_p^{N_p}}_{\text{point cloud features}}, \; \underbrace{\boldsymbol{h}_m^1, \dots, \boldsymbol{h}_m^{t-1}}_{\text{previous molecule features}} \Big). \tag{7}$$

**Inference** As shown in Fig. 3, we feed the conditioned point cloud into EDMolGPT and generate the molecular sequence $\mathcal{M}$ in an autoregressive, token-by-token manner. For FSMILES tokens $\widehat{a}_m^i$ and relative geometric tokens $\widehat{l}_m^i, \widehat{\theta}_m^i, \widehat{\phi}_m^i$, we apply temperature sampling (Radford et al., 2019) to draw predictions from the model's output distribution. However, directly sampling discretized 3D coordinates $\widehat{v}_m^i$ from the entire spatial domain often results in unrealistic or geometrically distorted molecular structures. To address this issue, we exploit the predicted relative geometric features to restrict the sampling space for $\boldsymbol{v}_m^i$. Specifically, given the three previously generated atom positions $\boldsymbol{v}_m^{i-1}, \boldsymbol{v}_m^{i-2}, \boldsymbol{v}_m^{i-3}$, and predicted $(l_m^i, \theta_m^i, \phi_m^i)$, we recover the continuous bond length, bond angle, and dihedral angle from their discretized representations. These quantities uniquely define a local reference frame, within which the feasible set of $\boldsymbol{v}_m^i$ lies on a spherical surface parameterized by $(l_m^i, \theta_m^i, \phi_m^i)$. The model then samples $\boldsymbol{v}_m^i$ exclusively from this constrained space, ensuring geometric consistency with the previously generated atoms while significantly reducing the search space and improving the efficiency and stability of autoregressive inference. More details about how to apply relative distance during inference are in the Appendix Sec. C

# 4 EXPERIMENT

## 4.1 SETTINGS

**Datasets** Our EDMolGPT model is trained on publicly available datasets[1], which include approximately eight million molecules along with their corresponding structural information. To improve data quality, we further filter the dataset using the Quantitative Estimate of Drug-likeness (QED) and the Synthetic Accessibility Score (SAS), resulting in a curated set of approximately two million molecules. For point cloud generation, we set $d_{\min} = 3.5$Å for each molecule. All point clouds are standardized to contain $N_p = 199$ points, which we find to offer a favorable trade-off between performance and computational efficiency.

For evaluation, we adopt the widely used DUD-E (Mysinger et al., 2012) benchmark dataset, which contains 101 receptors and their corresponding binding molecules. (Traditional DUD-E contains 102 targets. But following previous works (Feng et al., 2024), the target with the PDB ID 2H7L in the DUD-E dataset was excluded as it is listed as an obsolete entry in the PDB.) Following the same procedure used for pre-training data, we extract point clouds for each binding molecule and use them as conditional inputs during inference. For each receptor, we generate 1000 molecules for inference. We also give a more detailed discussion about distributions between the training data and inference data in Appendix Sec. E.1.

**Training/Evaluation Details** EDMolGPT is trained following the general setup of GPT-medium with a 24-layer Transformer backbone. We optimize the model using the AdamW (Loshchilov & Hutter, 2017) optimizer with a learning rate of $1 \times 10^{-5}$ and apply a warm-up schedule for the first 1000 steps before decaying according to a cosine schedule. The training is conducted with a batch size of 96 for 100 epochs. All experiments are performed on two NVIDIA A40 GPUs. For inference, we set the temperature $T = 0.7$ to scale the sampling distribution and make token prediction. More details about the architecture are in Appendix Sec. D.2.

**Evaluation Metrics** We compare EDMolGPT with previous methods utilizing ED: ECloudGen (Zhang et al., 2024) and ED2Mol (Li et al., 2025). For a fair comparison, we use the electron desnity drived from pre-existing binder to conduct experiments on ECloudGen, denoted as ECloudGen†. For reference, we report results on experimentally validated bioactive ligands, denoted as Reference. We also compare our method with SBDD methods: Pocket2Mol (Peng et al., 2022), TargetDiff (Guan et al., 2023b), Lingo3DMol (Feng et al., 2024), and MolCRAFT (Qu et al., 2024). We evaluate all methods from four complementary perspectives:

**(1)** Bioactive Molecule Recovery: The percentage of targets for which the generated compounds are similar to known active compounds, as measured by the Tanimoto similarity (Bajusz et al., 2015) of ECFP4 fingerprints (Rogers & Hahn, 2010). If at least one molecule generated by the method satisfies ECFP_TS > 0.5, we consider the corresponding active compound for that receptor as successfully recovered. We report the overall recovery rate across all pockets.

**(2)** Binding Affinity: We evaluate the binding affinity of generated ligands using GlideSP (Friesner et al., 2004) under two settings. In the *min-in-place* protocol, the generated conformation is directly minimized in the original binding pocket, preserving its pose, whereas *redocking* allows full flexible docking and repositioning of the ligand. Lower scores correspond to stronger predicted binding. To further assess alignment quality, we report the proportion of cases where *min-in-place* achieves a lower score than *redocking*, indicating that the generated conformations are already close to favorable binding modes and require little adjustment during docking. Glide is chosen for binding pose evaluation due to its wide adoption, strong ability to enrich active compounds, and frequent use as a reliable baseline in scoring function studies (Su et al., 2018; Shen et al., 2021).

**(3)** Conformational Stability: Assessed using the strain energy distribution of generated conformers. We evaluate via the commonly used Posecheck (Harris et al., 2023). We report the 25%, 50%, and 75% quantiles to reflect geometric reliability of the molecules.

**(4)** Molecular Properties: Evaluated using multiple criteria: drug-likeness (QED ↑), synthetic accessibility (SAS ↓), and average molecular weight. These metrics together capture the practicality

---

[1]Data available at: `http://data.aicnic.cn/dms-html/dataset_detail.html?id=848`

Table 1: The results with Binding Affinity, Conformation Stability (Conf. Stability), and Bioactive Molecule Recovery (Bioactive Mol. Recov. ), comparison across different methods. ↓ and ↑ indicate larger/smaller is better. Note: Min. < Re. denotes Min-in-place < Redocking. * *ED2Mol explicitly applies refinement utilizing external force fields before outputting molecules, a strategy not adopted by other models. This approach directly mitigates strain energy within the generated molecules.*

| Methods | Bioactive Mol. Recov. | Binding Affinity | | | Conf. Stability | | |
|---|---|---|---|---|---|---|---|
| | ECFP4_TS > 0.5 ↑ Ratio | Min-in-place ↓ Average | Redocking ↓ Average | Min. < Re. ↑ Ratio | Strain Energy ↓ | | |
| | | | | | 25% | 50% | 75% |
| Pocket2Mol | 8% | -6.7 | -7.5 | 17.9% | 69 | 126 | 230 |
| TargetDiff | 3% | -6.2 | -7.0 | 15.2% | 139 | 313 | 643 |
| Lingo3DMol | 33% | -6.8 | **-7.8** | 12.0% | 13 | 40 | 177 |
| MolCRAFT | 17% | -6.1 | -6.9 | 20.1% | 20 | 47 | 102 |
| ED2Mol | 3% | -5.22 | -6.15 | 7.4% | 7* | 24* | 57* |
| ECloudGen | 6% | - | - | - | - | - | - |
| ECloudGen† | 33% | - | -6.68 | - | - | - | - |
| EDMolGPT | **41%** | **-6.92** | -7.18 | **37%** | 33 | 69 | 194 |
| Reference | - | -7.93 | -7.93 | - | - | - | - |

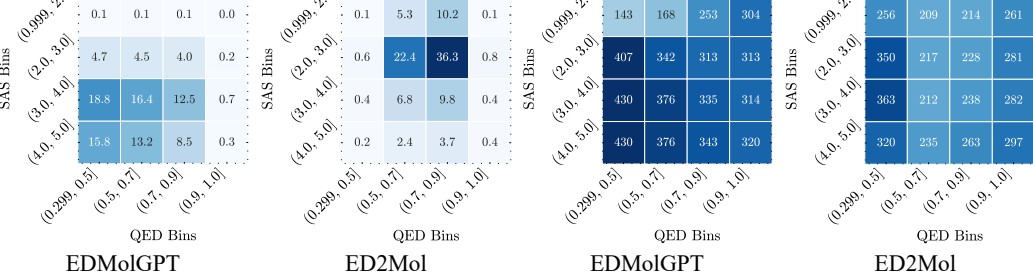

(a) Percentage of Samples by QED and SAS Bins     (b) Average Molecule Weight by QED and SAS Bins

Figure 4: The comparison between ED2Mol and EDMolGPT on QED, SAS, and Molecule Weight. We split QED and SAS into several bins and report the (a) Percentage of Samples by QED and SAS Bins and (b) Average Molecule Weight by QED and SAS Bins.

and overall quality of the generated molecules. Considering that observing one indicator alone is not very meaningful, we analyze the three indicators together in the following experiments.

## 4.2 MAIN RESULTS OF EDMOLGPT

**Results on Bioactive Molecule Recovery**    DUD-E, the dataset used for model evaluation, critically includes over 200 experimentally validated active ligands with measured affinities per target. This feature enables a direct comparison of AI-generated molecules against known active compounds, thereby mitigating the limitations of relying solely on purely computational assessments. Consequently, the Bioactive Molecule Recovery metric becomes an important metric for evaluating molecule generation models (Liu et al., 2024). On this crucial metric (Tab. 1), EDMolGPT achieves the highest recovery ratio among other methods, successfully reproducing bioactive molecules for 41% of the targets. This result implies that nearly half of the targets can be matched with generated compounds that are structurally similar (ECFP4_TS > 0.5) to known actives, highlighting the practical relevance of our method. The high recovery rate indicates that molecules generated by EDMolGPT are not only computationally favorable but also exhibit real biological activity.

**Results on Binding Affinity** As shown in Tab. 1, EDMolGPT achieves the lowest average Glide score in the min-in-place setting ($-6.92$), indicating that EDMolGPT-generated molecules adopt more favorable conformations for pocket-binding than those produced by other models. Beyond this absolute measure of binding quality, we also employed the widely adopted relative metric, the Min. < Re. ratio, to assess the quality of the generated binding conformations. This metric compares the pocket-binding quality of the generated conformation against the conformation obtained through classical force-field sampling. EDMolGPT demonstrated that 37% of its generated molecules exhibited binding modes superior to their force-field sampled counterparts, a performance that surpassed

other models. Collectively, these results demonstrate that EDMolGPT generates ligands with notably better pocket-binding modes compared to baseline models.

**Results on Conformational Stability** As shown in Tab. 1, we evaluate the conformational stability through strain energy analysis. While ED2Mol employs Qscore-guided (Terwilliger et al., 2006) fragment placement, which penalizes deviations from ideal bond geometries and applies force-field refinement (smina (Ding et al., 2022a)) during generation, EDMolGPT does not employ any post-processing, yet still achieves comparable quality. Specifically, the strain energies of EDMolGPT-generated molecules are 33, 69, and 194 kcal/mol at the 25%, 50%, and 75% quantiles, which are on par with those of Pocket2Mol and Lingo3dMol. These results indicate that EDMolGPT encodes 3D structural constraints during generation, producing conformations with competitive stability.

**Results on Properties** As shown in Tab. 2, we compare QED, SAS, and molecular weight across different methods. EDMolGPT achieves a strong balance across all three metrics, generating molecules with competitive QED and SAS values while maintaining molecular weights close to the Reference ligands—the ground-truth holo binders. This suggests that EDMolGPT produces chemically plausible molecules that reflect the size and complexity of real bioactive compounds. In contrast, some baselines, such as ED2Mol and ECloudGen, achieve favorable QED/SAS scores by generating smaller, simpler molecules or producing heavier but less chemically favorable structures.

Table 2: Results on QED, SAS, and molecule weight.

| Methods | QED | SAS | Mol. Weight |
|---|---|---|---|
| Pocket2Mol | 0.56 | 3.5 | 386 |
| TargetDiff | 0.60 | 4.0 | 299 |
| Lingo3DMol | 0.59 | 3.1 | 348 |
| MolCRAFT | 0.51 | 3.81 | 285 |
| ED2Mol | 0.73 | 3.9 | 234 |
| ECloudGen | 0.66 | 2.9 | 213 |
| ECloudGen† | 0.73 | 2.9 | 326 |
| EDMolGPT | 0.57 | 3.79 | 385 |
| Reference | 0.46 | 3.6 | 438 |

We also present a more detailed comparison between ED2Mol and EDMolGPT in Fig. 4. While ED2Mol attains higher QED and lower SAS, its molecules are generally much smaller, indicating that its apparent advantages stem from producing simpler structures rather than diverse, drug-like candidates, which limits its coverage of the chemical space relevant for realistic drug design.

**Results on cryo-EM-derived electron density** As shown in Tab. 3, we present the results of our model using cryo-EM-derived electron density. Our method is capable of generating valid molecules even under cryo-EM-derived densities, demonstrating its generalizability beyond purely computational data. However, since the model was trained on electron densities obtained via the FFT-cutoff and inverse FFT pipeline, the performance on cryo-EM-derived densities is slightly lower according to the Min-in-place metric. This difference reflects the distributional shift between training and cryo-EM data rather than a fundamental limitation of the model, and nevertheless demonstrates that our approach can generate valid molecules from experimental density maps to a reasonable extent.

Table 3: Results on cryo-EM-derived electron density.

| | Min-in-place |
|---|---|
| cryo-EM-derived | -5.90 |
| Ours | -6.92 |

### 4.3 ABLATION STUDIES

**Ablations studies on resolution** $d_{\min}$ To robustly enable our model to explore a broader chemical space and generate diverse scaffolds, differing significantly from the reference ligand while still being guided by the binding environment, we utilize low-resolution ED point clouds of reference binders. In our implementation, these low-resolution ED point clouds are achieved through two controls: the diffraction resolution ($d_{\min}$), which determines the coarseness of the electron density, and the ED grid resolution, governed by the number of sampling points ($N_p$), which dictates the sparsity of the point cloud. Crucially, our ablation studies on diffraction resolutions (Tab. 4) indicate that $N_p$ plays a more significant role in constructing this low-resolution representation. Specifically, setting $N_p = 199$ consistently generates ED point cloud representations that effectively mitigate bias associated with the reference binders from which the low-resolution ED point clouds are derived, irrespective of the chosen diffraction resolution. This is quantitatively supported by the consistently low Tanimoto similarity (typically $< 0.2$) between the generated molecules and the initial reference binders, as shown in Tab. 4 Div metric. Furthermore, varying the diffraction resolution does not introduce notable changes to the evaluation metrics.

**Ablation studies on temperature** Another key hyper-parameter in EDMolGPT is the sampling temperature $T$, which controls the randomness of the autoregressive generation process. Lower temperatures tend to produce more deterministic outputs, while higher temperatures encourage greater diversity but can introduce noisier conformations. As shown in

Table 4: Ablation studies on temperature and resolution. Div denotes the average score of ECFP_TS similarity between generated molecules and reference ligands, reflecting the structural diversity of the generated scaffolds.

| $d_{\min}$ | $T$ | Min-in-place | Redocking | Min. < Re. | Recov. | Div |
|---|---|---|---|---|---|---|
| 1.5Å | 0.7 | -6.94 | -7.12 | 33% | 46% | 0.186 |
| | 1.2 | -6.90 | -7.21 | 36% | 44% | 0.178 |
| 3.5Å | 0.7 | -6.92 | -7.18 | 37% | 41% | 0.184 |
| | 1.2 | -6.91 | -7.17 | 37% | 41% | 0.176 |

Tab. 4, increasing $T$ from 0.7 to 1.2 has a more noticeable impact on the Div score, which decreases slightly, indicating a modest reduction in scaffold diversity. Other metrics change only marginally: redocking scores improve slightly and the fraction of cases where the redocking score is lower than the minimum-in-place score increases, suggesting better pocket alignment, while the recovery rate drops modestly. These results show that sampling temperature significantly influences the balance between diversity and structural consistency in generation.

**Ablation studies on $N_p$ and pharmacophore labels $c_p$** As shown in Tab. 5, we perform ablation studies on $N_p$ and the use of pharmacophore labels $c_p$, which are key hyperparameters for modeling the electron density. Increasing $N_p$ provides a more detailed description of the positive electron-density patterns, improving the Min-in-place score while slightly reducing diversity. Conversely, removing pharmacophore labels $c_p$ relaxes geometric constraints,

Table 5: Ablation results on $N_p$ and pharmacophore labels.

| | Min-in-place | Div |
|---|---|---|
| $N_p = 100$ | -6.46 | 0.15 |
| $N_p = 300$ | -7.22 | 0.20 |
| w/o $c_p$ | -6.15 | 0.09 |

increasing diversity but lowering Min-in-place scores. These results illustrate the expected trade-offs and confirm that both $N_p$ and pharmacophore guidance play important roles in ensuring the quality and structural fidelity of the generated molecules.

## 5 DISCUSSION

Unlike conventional structure-based generative approaches that rely on rigid pocket geometries, EDMolGPT leverages low-resolution electron density from pre-existing binders to guide generation. This makes the method applicable in two scenarios: (i) when a protein is solved in holo-state with a binder (cofactors or tool compounds), common in early discovery, and (ii) when the pocket structure is unavailable, as conformations of known actives can be obtained by molecular dynamics and converted into density. Prior studies show that dominant solution conformations of active compounds often resemble their bound states, supporting this strategy (Tong & Zhao, 2021; Gu et al., 2021). While related to Ligand-Based Drug Design (LBDD), EDMolGPT requires only one reference binder or compound rather than an activity series, and directly enables 3D generation beyond the 2D or 1D fingerprints used in LBDD. Given that over 96% of clinical trials focus on targets with known ligand or target information (Vasan et al., 2023), EDMolGPT advances pocket-structure-based methods, offering a robust solution for a wide range of real-world discovery scenarios. Further discussion of these application scenarios is provided in Appendix Sec. A.

## 6 CONCLUSION

In this work, we present EDMolGPT, a novel decoder-only autoregressive framework for 3D drug design that leverages electron-density–derived point clouds as conditioning signals. By sampling low-resolution electron density from an existing binder instead of relying on rigid pocket representations, our approach flexibly describes the binding environment, enabling generation of ligands with chemically plausible conformations. Experiments on over 100 DUD-E targets show EDMolGPT outperforms existing structure-based generative methods in both 3D and 2D chemical spaces, with improved binding modes and higher recovery rates of bioactive molecules. We believe EDMolGPT offers a powerful paradigm to advance current technologies in drug design by opening new opportunities for efficient and biologically meaningful molecule generation.

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

# SUPPLEMENTARY MATERIAL

## OVERVIEW

This appendix presents comprehensive experimental details, evaluation details, and more visualization results. The content is organized into five main sections:

- Sec. A discusses the practical significance of electron density–based generation and contrasts it with pocket-based generation.

- Sec. B.1, Sec. B.2, and Sec. B.3 provide further details on FSMILES, discretized 3D coordinates, and relative distances, supplementing the main text with specifications of the molecular input format.

- Sec. D.1, Sec. D.2 and Sec. D.3 provide additional details on the DUD-E dataset, the hyperparameters of EDMolGPT, and more explanation for Bioactive Molecule Recovery metric.

- Sec. C explains how relative distance is employed to constrain the prediction of discretized 3D coordinates.

- Sec. E.1 examines the differences in data distributions between the public dataset used to train EDMolGPT and the DUD-E dataset, while Sec. E.2 presents additional visualization results. Sec. E.3 presents the computational efficieny between different methods.

## A  DISCUSSION

Unlike previous structure-based generative approaches that explicitly rely on the precise 3D geometry of the protein pocket and often operate under a rigid pocket assumption, EDMolGPT leverages low-resolution electron density, derived from pre-existing binders within the pocket, to guide molecular generation.

The broad applicability of our method in practical drug discovery scenarios is evident from two key perspectives. Firstly, it is highly applicable whenever a protein structure has been solved with a binder in the pocket, a common occurrence in the early stages of drug discovery. Observing a binder within a potential binding site is frequently a prerequisite for confirming its designation as a 'pocket' and for establishing its holo-state for subsequent drug screening or design. This holds true regardless of whether the binder is a natural cofactor (e.g., NADPH or ADP) or a reference tool compound, all of which our model can utilize. Secondly, our method's versatility extends to scenarios where the binding pocket structure itself has not been experimentally solved. In such instances, if an active compound is known, its major conformations in aqueous solution can be determined through molecular dynamics simulations. These computationally derived conformations can then be utilized for the construction of low-resolution electron density, which subsequently fuels our method. This approach is supported by prior research indicating that highly active compounds tend to exhibit minimal differences between their dominant solution-phase conformations and their bound conformations, thereby incurring low strain energy upon binding (Tong & Zhao, 2021; Gu et al., 2021). While our method, to some extent, shares conceptual similarities with Ligand-Based Drug Design (LBDD), it offers broader applicability and distinct advantages. Current LBDD techniques typically require a series of ligands with a wide range of activities to establish Structure-Activity Relationships (SAR) and primarily focus on 2D molecular generation based on molecular fingerprints. In contrast, EDMolGPT offers a much simplified input requirement and provides unique strengths in the 3D generation space.

Considering that over 96% of clinical trials in practical drug discovery pipelines focus on previously studied targets with either known ligand or target information (Vasan et al., 2023), EDMolGPT is well-positioned to address the majority of real-world drug discovery cases, offering a complementary paradigm to conventional pocket-structure-based design methods.

## B    REPRESENTATION OF MOLECULE

### B.1    FSMILES

The original FSMILES was developed for the autoregressive task in Lingo3DMol. The pre-defined token vocabulary is shown as follows:

```
fsmiles_list = [
            "pad_0", "start_0", "end_0", "sep_0",
            "C_0", "C_5", "C_6", "C_10", "C_11", "C_12",
            "c_0", "c_5", "c_6", "c_10", "c_11", "c_12",
            "N_0", "N_5", "N_6", "N_10", "N_11", "N_12",
            "n_0", "n_5", "n_6", "n_10", "n_11", "n_12",
            "S_0",
            "s_0", "s_5", "s_6", "s_10", "s_11", "s_12",
            "O_0", "O_5", "O_6", "O_10", "O_11", "O_12",
            "o_0", "o_5", "o_6", "+_0", "o_11", "o_12",
            "F_0",
            "Cl_0",
            "[nH]_0", "[nH]_5", "[nH]_6",
            "[nH]_10", "[nH]_11", "[nH]_12",
            "Br_0",
            "/_0", "\\_0", "@_0", "@@_0", "H_0",
            "1_0", "2_0", "3_0", "4_0", "5_0", "6_0",
            "#_0", "=_0", "-_0", "(_0", ")_0",
            "[_0", "]_0", "[*]_0","([*])_0"
        ].
```

In FSMILES, the tokens pad_0, start_0, end_0, and sep_0 denote the padding token, the start-of-molecule marker, the end-of-molecule marker, and the fragment separator, respectively. However, the current FSMILES implementation, which is based on BRICS (Degen et al., 2008), often produces an excessive number of small, highly fragmented substructures. To address this limitation, we refine the FSMILES decomposition process by introducing a fragment-consolidation strategy. Specifically, we preserve any bond whose cleavage would generate fragments containing fewer than three atoms, thereby mitigating over-fragmentation. As illustrated in Fig. 5, our method prevents unnecessary segmentation. For example, in the left panel, the hydroxyl group (-OH) is cleaved in the original FSMILES decomposition, whereas our approach preserves it, resulting in a more chemically meaningful fragment.

For the better understanding of FSMILS, we also visualize the density of fragment atom count and the density of fragment molecular weight. As shown in Fig. 6, the majority of fragments contain a moderate number of atoms, and the molecular weights are concentrated within a reasonable range, indicating that the fragment decomposition produces chemically meaningful substructures. These distributions suggest that our fragmentation strategy successfully balances the granularity of molecular breakdown with the preservation of structurally relevant motifs, avoiding excessive over-fragmentation while maintaining fragments suitable for downstream modeling.

### B.2    DISCRETIZED 3D COORDINATES

We provide further clarifications regarding the discretization scheme described in the main text.

First, the choice of resolution $\sigma = 0.1$ Å reflects a trade-off between geometric fidelity and vocabulary size. With this setting, the maximum quantization error per coordinate dimension is 0.05 Å, which is negligible compared to the typical bond length in organic molecules ($\sim 1.2$–1.5 Å). Thus, the discretized representation is sufficiently accurate to preserve chemically meaningful structures.

Second, the positive shift applied after discretization ensures that all coordinates $\widehat{v}_m^i$ lie within the range of non-negative integers. This design is important because it allows us to directly map each integer triplet to a unique token in the model vocabulary without introducing negative indices, thereby simplifying both tokenization and embedding lookups.

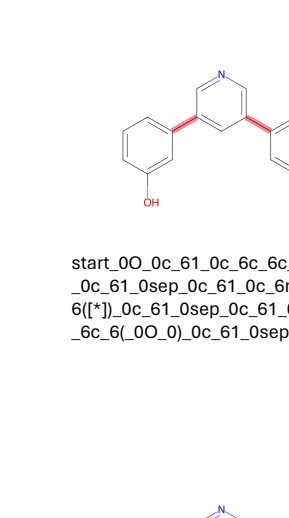
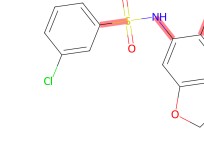

start_0O_0c_61_0c_6c_6c_6c_6([*])
_0c_61_0sep_0c_61_0c_6n_6c_6c_
6([*])_0c_61_0sep_0c_61_0c_6c_6c
_6c_6(_0O_0)_0c_61_0sep_0end_0

start_0O_0=_0S_0([*])_0(_0=_0O_0)
_0N_0[*]_0sep_0c_61_0c_6c_112_0
c_11(_0c_6c_61_0[*]_0)_0O_5C_5O
_52_0sep_0c_51_0n_5c_5c_5o_51_
0sep_0c_61_0c_6c_6c_6c_6(_0Cl_0
)_0c_61_0sep_0end_0

start_0C_0C_0([*])_0C_0sep_0c_61
_0c_6c_6([*])_0n_6c_122_0c_6c_6c
_6c_6c_121_02_0sep_0N_0N_0C_0
(_0=_0O_0)_0N_0[*]_0sep_0c_61_0
c_6c_6(_0Cl_0)_0c_6c_6(_0Cl_0)_0
c_61_0sep_0end_0

(a) FSMILES

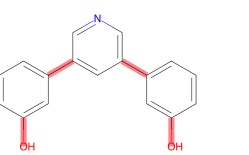
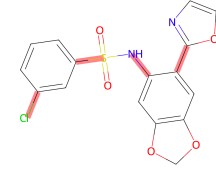
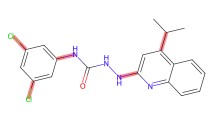

start_0O_0[*]_0sep_0c_61_0c_6c_6
c_6c_6([*])_0c_61_0sep_0c_61_0c_
6n_6c_6c_6([*])_0c_61_0sep_0c_61
_0c_6c_6c_6c_6([*])_0c_61_0sep_0
O_0sep_0end_0

start_0O_0=_0S_0([*])_0(_0=_0O_0)
_0N_0[*]_0sep_0c_61_0c_6c_112_0
c_11(_0c_6c_61_0[*]_0)_0O_5C_5O
_52_0sep_0c_51_0n_5c_5c_5o_51_
0sep_0c_61_0c_6c_6c_6c_6([*])_0c
_61_0sep_0Cl_0sep_0end_0

start_0C_0C_0([*])_0C_0sep_0c_61
_0c_6c_6([*])_0n_6c_122_0c_6c_6c
_6c_6c_121_02_0sep_0N_0N_0C_0
(_0=_0O_0)_0N_0[*]_0sep_0c_61_0
c_6c_6([*])_0c_6c_6([*])_0c_61_0se
p_0Cl_0sep_0Cl_0sep_0end_0

(b) Ours

Figure 5: The comparison between (a) FSMILES and (b) Ours. We highlight the cut bonds in red, and the tokenized result is marked below.

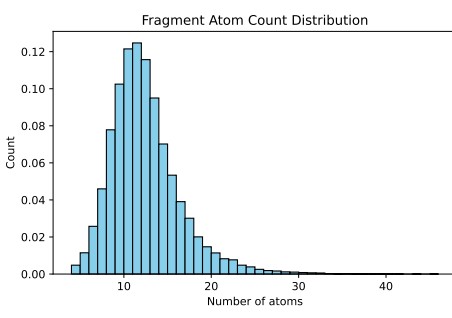
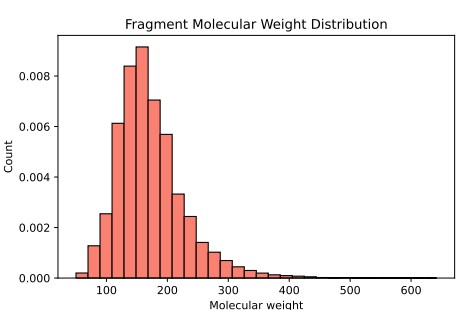

(a) The density of fragment atom count

(b) The density of fragment molecular weight

Figure 6: The visualization results on (a) Fragment Atom Count Distribution and (b) Fragment Molecular Weight Distribution.

Third, although discretization maps continuous space into a bounded integer lattice, the autoregressive model does not rely solely on absolute coordinate tokens. Instead, the generation process is conditioned on relative geometric features (bond length $l$, bond angle $\theta$, and dihedral angle $\phi$), which act as local structural constraints during inference. This hybrid formulation mitigates the potential artifacts of quantization by guiding the coordinate recovery step toward chemically valid regions.

Finally, both molecular coordinates and auxiliary point clouds are discretized into the same shifted lattice space. This alignment enables us to encode heterogeneous geometric information (e.g.,

atomic positions and electron-density-derived point clouds) within a unified tokenization frame-work, which greatly facilitates multimodal training.

## B.3 RELATIVE DISTANCE

To determine the reference atoms required for autoregressive coordinate generation, we design a procedure to trace the ancestral nodes of each token in the molecular sequence $\mathcal{M} = \{a_m^1, a_m^2, \ldots, a_m^n\}$. For a given step $i$, we define three levels of ancestor indices:

$$r_1(i), \quad r_2(i) = r_1(r_1(i)), \quad r_3(i) = r_1(r_2(i)), \tag{8}$$

where $r_1(i)$ denotes the *first-order ancestor*, $r_2(i)$ the *second-order ancestor*, and $r_3(i)$ the *third-order ancestor* of token $a_m^i$.

The search for $r_1(i)$ is performed by traversing the sequence backward from step $i$: (1) if $a_m^i$ is an atom token, $r_1(i)$ is assigned to the nearest preceding atom token; (2) if $a_m^i$ is a non-element symbol (e.g., branch markers "(" and ")"), we recursively skip bracketed fragments while ensuring valid pairing of parentheses, thus locating the chemically valid attachment point of the current sub-structure; (3) if a separator token sep_0 is encountered, the search crosses fragment boundaries and jumps to the most recent star marker $[\cdot]$ that denotes a fragment connection point.

Once $r_1(i)$ is determined, higher-order ancestors are obtained recursively as

$$r_2(i) = r_1(r_1(i)), \qquad r_3(i) = r_1(r_2(i)). \tag{9}$$

These indices provide the hierarchical reference atoms for step $i$, which correspond to the spatial coordinates

$$\boldsymbol{v}_m^{i-1} = \boldsymbol{v}_{r_1(i)}, \quad \boldsymbol{v}_m^{i-2} = \boldsymbol{v}_{r_2(i)}, \quad \boldsymbol{v}_m^{i-3} = \boldsymbol{v}_{r_3(i)}. \tag{10}$$

In this way, the algorithm ensures that each new atom position $\boldsymbol{v}_m^i$ is generated with respect to three previously defined reference atoms, enabling consistent computation of bond length, bond angle, and dihedral angle during molecular construction.

## C DETAILS ABOUT INFERENCE

During inference, we feed the conditioned point cloud into EDMolGPT and generate the molecular sequence $\mathcal{M}$ in an autoregressive, token-by-token manner. For FSMILES tokens $\widehat{a}_m^i$ and relative geometric tokens $\widehat{l}_m^i, \widehat{\theta}_m^i, \widehat{\phi}_m^i$, we apply temperature sampling (Radford et al., 2019) to draw predictions from the model's output distribution. However, directly sampling discretized 3D coordinates $\widehat{\boldsymbol{v}}_m^i$ from the entire spatial domain often results in unrealistic or geometrically distorted molecular structures. To address this issue, we exploit the predicted relative geometric features to restrict the sampling space for $\boldsymbol{v}_m^i$.

Specifically, given the three previously generated atom positions $\boldsymbol{v}_m^{i-1}, \boldsymbol{v}_m^{i-2}, \boldsymbol{v}_m^{i-3}$, and the predicted discretized features $(\widehat{l}_m^i, \widehat{\theta}_m^i, \widehat{\phi}_m^i)$, we first recover the corresponding continuous values according to the discretization rule (cf. Eq. 5):

$$l_m^i = \widehat{l}_m^i \cdot \sigma, \quad \theta_m^i = \widehat{\theta}_m^i \cdot 10°, \quad \phi_m^i = \widehat{\phi}_m^i \cdot 10°. \tag{11}$$

Instead of enforcing these values deterministically, we introduce tolerance intervals:

$$l_m^i \in [l_m^i - \delta_l, l_m^i + \delta_l,], \theta_m^i \in [\theta_m^i - \delta_\theta, \theta_m^i + \delta_\theta], \phi_m^i \in [\phi_m^i - \delta_\phi, \phi_m^i + \delta_\phi], \tag{12}$$

where $\delta_l, \delta_\theta$, and $\delta_\phi$ control the flexibility of bond length, bond angle, and dihedral angle, respectively (empirically, $\delta_l \approx 0.1\text{Å}, \delta_\theta \approx 10°, \delta_\phi \approx 10°$).

Accordingly, the feasible region of the next atom coordinate $\boldsymbol{v}_m^i$ is constrained as

$$\boldsymbol{v}_m^i \in \Big\{ \boldsymbol{v} \in \mathbb{R}^3 \Big| |\boldsymbol{v} - \boldsymbol{v}_m^{i-1}| \in [l_m^i - \delta_l, l_m^i + \delta_l],$$
$$\theta(\boldsymbol{v}) \in [\theta_m^i - \delta_\theta, \theta_m^i + \delta_\theta], \tag{13}$$
$$\phi(\boldsymbol{v}) \in [\phi_m^i - \delta_\phi, \phi_m^i + \delta_\phi] \Big\}.$$

This procedure constrains $\boldsymbol{v}_m^i$ to a narrow spherical patch determined by the predicted local structure, thereby ensuring geometric consistency with the previously generated atoms while allowing moderate flexibility to account for discretization errors and model uncertainty.

Table 6: Hyperparameter settings for our EDMolGPT based model.

| Hyperparameter | Value |
|---|---|
| input_vocab_size | 300 |
| input_dist_size | 300 |
| num_bond_ang | 200 |
| num_bond_leng | 200 |
| num_dih_ang | 200 |
| n_layer | 24 |
| n_embd | 1024 |
| n_ctx (= n_positions) | 1024 |
| n_head | 16 |
| activation_function | gelu_new |
| resid_pdrop | 0.1 |
| embd_pdrop | 0.1 |
| attn_pdrop | 0.1 |
| layer_norm_epsilon | 1e-5 |
| initializer_range | 0.02 |

## D  EXPERIMENTAL SETTING

### D.1  DUD-E DATASET

The DUD-E (Directory of Useful Decoys: Enhanced) dataset is a large-scale benchmark designed for the development and evaluation of virtual screening algorithms. It is an enhanced version of the original DUD dataset, constructed by the Shoichet Laboratory at the University of California, San Francisco (UCSF). DUD-E comprises 102 protein targets, spanning diverse families such as kinases, proteases, GPCRs, nuclear receptors, and ion channels, thereby covering a broad spectrum of pharmacologically relevant classes. For each target, the dataset provides a curated set of experimentally validated active ligands together with approximately 50 property-matched decoys per active compound. These decoys are selected to mimic the actives in terms of simple physicochemical descriptors (e.g., molecular weight, hydrogen bond donors/acceptors, logP), but are topologically distinct, making them unlikely to bind the target. This careful design allows DUD-E to reduce dataset bias and to more reliably evaluate the discriminative power of computational screening methods. Consequently, DUD-E has become a standard benchmark for assessing the performance of molecular docking, machine learning–based virtual screening, and structure-based drug design approaches.

### D.2  HYPER-PARAMETERS OF EDMOLGPT

As shown in Tab. 6, we summarize the hyperparameter settings of our EDMolGPT model. The vocabulary size for FSMILES tokens and coordinate tokens are specified by input_vocab_size and input_dist_size, both set to 300, which we found sufficiently large to accommodate current requirements while leaving room for future extensions. Similarly, the numbers of tokens representing bond lengths, bond angles, and dihedral angles (num_bond_leng, num_bond_ang, and num_dih_ang) are each set to 200, ensuring adequate coverage of structural variations with flexibility for expansion.

For the general architecture, we extend the GPT-2 backbone with n_layer = 24 transformer blocks, hidden dimension n_embd = 1024, context length n_ctx = 1024, and n_head = 16 attention heads, which provide higher model capacity suitable for molecular generation. The activation function is set to gelu_new (Hendrycks & Gimpel, 2016), while dropout is consistently applied at multiple levels (resid_pdrop, embd_pdrop, attn_pdrop, all 0.1) to mitigate overfitting. Other parameters, including the layer normalization epsilon (1e-5) and weight initialization range (0.02), follow the standard GPT-2 configuration to ensure stable optimization.

Overall, our design largely inherits the strengths of GPT-2 while incorporating domain-specific vocabulary extensions and scaling adjustments to better capture molecular structural information.

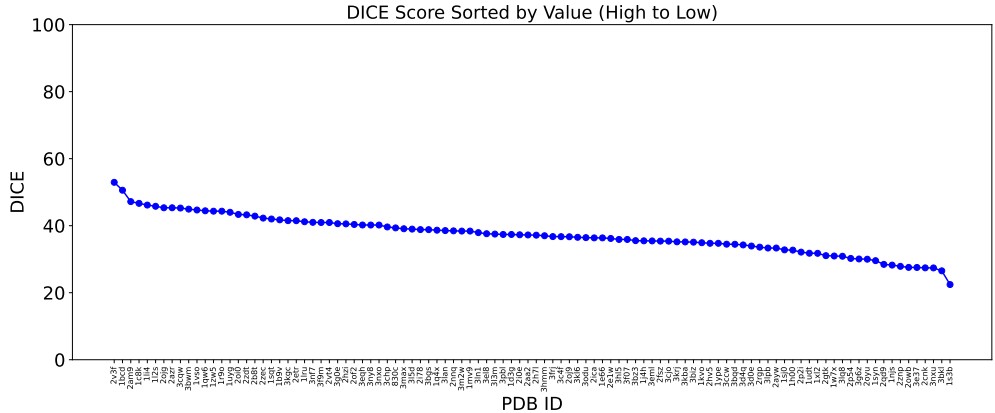

Figure 7: DICE similarity scores between DUD-E active ligands and their closest counterparts in the training dataset, sorted from high to low. Each point corresponds to a DUD-E ligand, with the horizontal axis indicating the PDB ID and the vertical axis showing the maximum DICE score identified in the training set. The results indicate that all maximum DICE scores remain below 60%.

### D.3 METRICS FOR BIOACTIVE MOLECULE RECOVERY

We use several metrics to evaluate the biological relevance and structural quality of our generated molecules. Among these, the ECFP4 Tanimoto Similarity (TS) is employed to specifically measure the ability of the model to generate molecules that exhibit high putative biological activity, which we detail here. ECFP4 (Extended Connectivity Fingerprints, diameter 4) is a widely accepted circular fingerprint in chemoinformatics that encodes the local chemical environment around each atom, capturing essential pharmacophoric features strongly correlated with bioactivity. We evaluate this metric by comparing our generated molecules against a set of already-validated active molecules corresponding to the target pockets, which are sourced from the established DUDE (Database of Useful Decoys: Enhanced) benchmark. A high ECFP4 TS score between a generated molecule and a known active molecule indicates that the generated compound successfully recapitulates or is structurally analogous to a proven active scaffold. This measure serves as a crucial chemoinformatics-based validation and effectively complements purely physical metrics like docking scores, allowing us to holistically assess that the generated output is not only stable in the pocket but also chemically plausible as a bioactive compound.

## E MORE EXPERIMENTAL RESULTS

### E.1 DISTRIBUTION ANALYSIS

To ensure that our method truly generates novel molecules rather than memorizing those from the training set, we conducted a distributional overlap analysis between the training and test data. The training set consists of approximately two million molecules curated from public databases with additional drug-likeness filtering. In contrast, the DUD-E dataset, used for evaluation, was independently processed into point cloud representations of active ligands. Since it is unclear whether any structural overlap exists between the DUD-E molecules and the training set, it is necessary to explicitly verify the degree of intersection. If a significant overlap were present, one could not rule out the possibility that our model merely recalls known molecules instead of generating meaningful new candidates.

To address this, we performed a nearest-neighbor retrieval experiment: for each active ligand point cloud from DUD-E, we searched the training set for the most similar point cloud according to the DICE coefficient, and recorded the corresponding similarity scores. Visualization of the results (Fig. 7) shows that the maximum DICE similarity never exceeds 60%. This low overlap confirms that the test set structures are not contained in the training data, thereby validating that our model's outputs represent genuine generation rather than memorization.

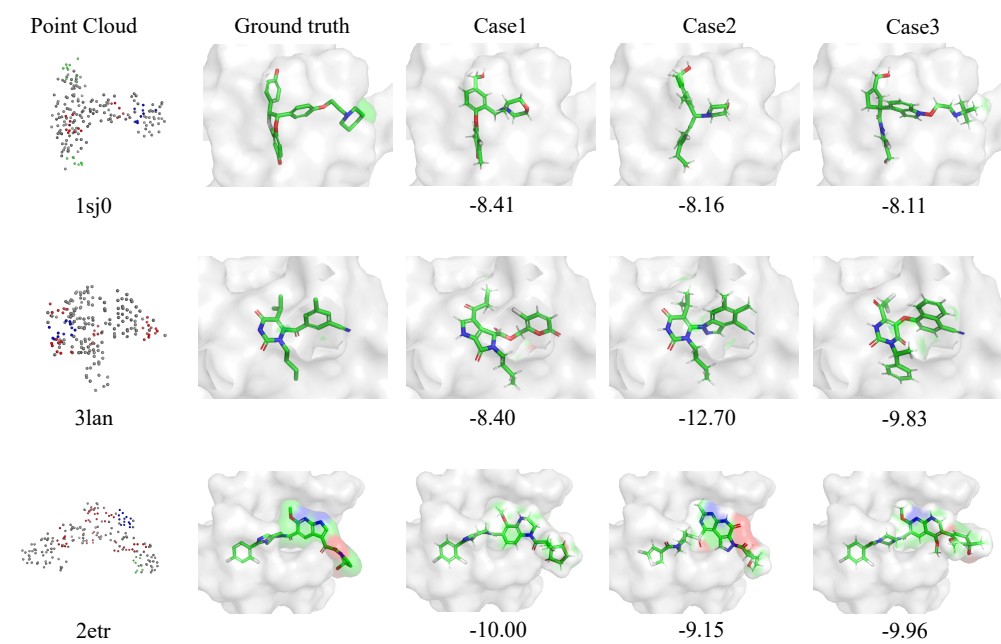

Figure 8: Visualization of three protein–ligand complexes with PDB IDs 1sj0, 3lan, and 2etr. The first column shows the point cloud extracted from the electron density map. The second column presents the ground-truth ligand conformations within the corresponding protein pockets. The following three columns (Case 1–3) display ligands generated by our method, with the associated minimum in-place docking scores indicated below each case.

### E.2 VISUALIZATION OF GENERATION RESULTS

To further evaluate the effectiveness of our framework, we conducted visualization analyses on three representative protein–ligand complexes (1sj0, 3lan, and 2etr), as shown in Fig. 8. For each target, the electron density–derived point cloud was used as the conditioning input, and we compared the generated ligands against the experimentally determined ground truth. The visualizations demonstrate that our method produces ligands that align well with the spatial distribution of the point cloud, indicating that the generative process effectively captures the underlying geometric constraints of the binding pocket. In addition, the generated ligands consistently achieve favorable docking scores, often lower than those of the reference structures, suggesting strong binding compatibility and chemical plausibility. Importantly, across all three systems, our framework is able to generate multiple diverse ligand candidates while maintaining close agreement with the pocket environment. This combination of low docking scores and structural diversity highlights the rationality of our design and suggests that the method can reliably explore alternative binding modes without sacrificing physical or chemical feasibility.

### E.3 COMPUTATIONAL EFFICIENCY

As shown in Tab. 7, EDMolGPT utilizes an autoregressive GPT architecture conditioned by the electron density map, and achieves a competitive average generation speed of approximately 1.5 seconds per molecule. For comparison, we have compiled the reported or estimated generation speeds for several prominent SBDD models. Pocket2Mol, another autoregressive model, reports a highly optimized generation speed of approximately 0.45 seconds per molecule. For diffusion-based models like TargetDiff, the speed depends heavily on the number of sampling steps (TargetDiff uses 1000 steps); the multi-step nature of the diffusion process typically makes them significantly slower than highly optimized autoregressive models.

Table 7: Comparison of molecular generation speed across different models.

| Model | Architecture | Avg. Generation Time (s/molecule) |
|---|---|---|
| ED-GPT | Autoregressive | $\approx 1.5$ |
| Pocket2Mol | Autoregressive | $\approx 0.45$ |
| TargetDiff | Diffusion | $\approx 7$ |

## USE OF LLMS

The paper has been polished with the assistance of a large language model (LLM) to improve clarity and readability. All ideas, experiences, and research descriptions are my own, and the content accurately reflects my background and intentions.

