# OpenReview forum: "EDMolGPT: A Decoder-Only Framework for 3D Drug Design via Electron Density"
_ICLR.cc/2026/Conference — Submitted to ICLR 2026_

### Official Review · Reviewer_ApF4 · 2025-10-31

**Soundness:** 2
**Presentation:** 2
**Contribution:** 2
**Rating:** 2
**Confidence:** 4

**Summary:**

The paper proposes a drug design model based on the electron density representation of known ligands, employing an autoregressive Transformer architecture and introducing the FSMILES representation to achieve 3D molecular generation. The method extracts low-resolution point cloud representations via FFT and a high-frequency cutoff, and incorporates hydrogen-bonding information to provide a novel guidance mechanism for molecular generation.

Experiments are conducted on the DUD-E dataset to validate the approach on new ligands, and the Glide score is used as evaluation metrics, which better reflects real-world drug design scenarios.

However, the paper suffers from unclear representation, weak argumentation, insufficient baselines, and incomplete experimental analysis, all of which require further improvement and refinement.

**Strengths:**

- Method: This paper proposes a novel representation to guide 3D drug design, which integrates fragment, binding, and spatial information — all critical for real world application. The method also addresses key challenges in efficiency and scalability associated with spatial point cloud representations.

- Experiments: The study uses the DUD-E dataset as the test set and reports Glide scores, which provide more practically relevant and application-oriented evaluations compared to baselines that use CrossDocked datasets and Vina scores.

**Weaknesses:**

- Task: The reviewer argues that this work still essentially belongs to ligand-based drug design (LBDD), merely adopting a different form of conditional representation. Besides, the paper lacks controlled comparisons to demonstrate that the proposed representation is superior to those used in ED2Mol or ECloudGen.
- Presentation: Several main claims are not rigorously supported.
    - Error accumulation: The proposed point cloud representation itself still contains errors, which may arise from multiple sources — including binding pose, electron density calculation, h bond annotation, and discretization precision loss. Although the authors claim that their approach reduces error accumulation compared to two-stage methods, the representation they use is also error-prone, and the paper does not report any quantitative error analysis.
    - Flexibility: The claimed advantage of not requiring explicit pocket structures is actually a common feature of LBDD, rather than a unique contribution of this work. Moreover, since the point cloud is derived from a single ligand conformation, it cannot capture pocket flexibility.
    - Minor points: The paper lacks a clear demonstration or ablation of FSMILES effectiveness, and the biological activity metric ECFP4 Tanimoto similarity (TS) is not clearly explained.
    - Docking score: How is the initial binding pose determined when calculating min score? Can you provide the experiment affinity or docking scores for reference ligands?
- Method:
    - Sorting point clouds purely by coordinate order may disrupt spatial locality.
    - The paper does not explain how positional embeddings — a critical component of Transformer architectures — are implemented in this context.
- Experiment:
    - Table 1 does not report the average molecular weight of each model. ED2Mol tends to generate smaller molecules, which typically exhibit lower docking scores, lower strain energy, and higher drug-likeness. The paper only reports ED2Mol’s molecular weight, but ED2Mol performs weakly overall (Table 1). The authors should have compared against stronger baselines.
    - Table 1 also omits important metrics such as diversity and novelty. Furthermore, Figure 6 only verifies that there is no data leakage between training and test sets, but does not confirm that the generated molecules are novel relative to the training data. Intuitively, for certain small molecules, using 199 points may provide excessive information about the reference ligand, effectively making the ligand-based design task easier. The authors need to demonstrate that their method can generate molecules that are significantly different from the reference ligand, rather than merely reproducing or slightly modifying it.

**Questions:**

see weakness

---

> ### Author Response · Authors · 2025-11-22
>
> We thank the reviewer for the insightful and highly professional comments. We have addressed these concerns in the revised manuscript and highlighted the changes in blue. Below, we summarize your points in quotes, followed by our responses.
>
> > Task: The reviewer argues that this work still essentially belongs to ligand-based drug design (LBDD), merely adopting a different form of conditional representation. Besides, the paper lacks controlled comparisons to demonstrate that the proposed representation is superior to those used in ED2Mol or ECloudGen.
>
> ***response:*** We thank the reviewer for raising this point. Both ED2Mol and ECloudGen are also based on electron-density representations. We tested ECloudGen on the electron densities used in our application and found that it underperforms compared to our method across multiple metrics, including redocking scores and active molecule recovery, as shown in follows.
> |Method|	ECFP4\_TS $> 0.5$|Redocking|
> |-|-|-|
> EcloudGen | 33%|	-6.68
> EDMolGPT|41%|	-7.18
>
> For ED2Mol, we observed that it is difficult to apply the method on experimentally derived densities, as the ED2Mol-predicted densities differ substantially from real densities, preventing effective guidance for molecule generation. These experiments demonstrate that, under a fair comparison with appropriate electron-density inputs, our approach surpasses both ED2Mol and ECloudGen.
>
> > Error accumulation: The proposed point cloud representation itself still contains errors, which may arise from multiple sources — including binding pose, electron density calculation, h bond annotation, and discretization precision loss. Although the authors claim that their approach reduces error accumulation compared to two-stage methods, the representation they use is also error-prone, and the paper does not report any quantitative error analysis.
>
> ***response:*** We agree that point-cloud estimation inevitably introduces errors from multiple stages, such as binding pose prediction, coarse-grained ED estimation, hydrogen bond annotation, and grid discretization. These limitations are intrinsic to all approaches operating on predicted density rather than experimentally measured ED.
>
> However, our motivation is that existing two-stage pipelines suffer from compounded errors: the density distribution error propagates from voxel-based ED regression to subsequent structure representation. In contrast, our method bypasses the intermediate ED distribution field and directly models the interaction-relevant point cloud representation. Therefore, while the representation itself is not error-free, it avoids the secondary amplification of ED intensity misalignment, which we observed to be substantial in baseline approaches.
>
> To support this, we additionally visualize the intensity discrepancy between the true ED and the estimated density used in prior workflows. As shown in Fig.2 of the revised manuscript, the spatial pattern and magnitude of their ED mismatch indicate that relying on predicted ED intensity fields can mislead downstream interaction modeling. In comparison, our representation retains a more stable geometric correlation with the ligand-protein interface.
>
> We have clarified these error sources and added a supplementary quantitative analysis in the revised version.

---

> > ### Author Response · Authors · 2025-11-22
> >
> > > Flexibility: The claimed advantage of not requiring explicit pocket structures is actually a common feature of LBDD, rather than a unique contribution of this work. Moreover, since the point cloud is derived from a single ligand conformation, it cannot capture pocket flexibility.
> >
> > ***response:***  Traditional LBDD often relies on 2D fingerprints, pharmacophores, or similarity to a rigid input ligand. Our approach uses a unique, experimentally informed 3D spatial constraint as the condition for an autoregressive model.
> >
> > Regarding the assertion that deriving the ED from a single ligand conformation cannot capture pocket flexibility, we offer a more nuanced explanation that addresses this conceptual mismatch: While the ED is derived from the ligand's experimentally determined coordinates, we utilize this information as a high-fidelity proxy for the binding pocket's most critical spatial and geometric constraints. The ligand's conformation is the result of its instantaneous interaction with the pocket, inherently reflecting the boundaries of the site. Crucially, our methodology intentionally utilizes a low-resolution, coarse-grained representation of this electron density. This is a deliberate strategy to introduce flexibility into the system: the low-resolution nature smooths out the hyper-specific details of the single static conformation, preventing the generative model from being locked into a strictly rigid pocket assumption. Furthermore, this "soft" constraint grants the model sufficient degrees of freedom to explore alternative, low-energy conformations that still reside within the broader, permissible spatial envelope dictated by the coarse-grained ED. Thus, by converting the single ligand conformation into a low-resolution, flexible spatial constraint, our method effectively and computationally samples the transient flexibility of the binding site, which is conceptually aligned with the goal of generating novel molecules with diverse scaffolds.
> >
> > **We also provide a detailed example to further validate our claim. As shown in Fig.1 in the paper, biochemical assays confirm the activity of ligands bearing bulky substituents at this site, indicating that the binding site is conformationally flexible and not comprehensively represented by the static structure. Accordingly, the generation constrained by the static pocket fails to produce these active compounds. In contrast, our low-resolution ED–guided approach accommodates local conformational plasticity, enabling the successful generation of bulky, yet active, substituents.**
> >
> > > Minor points: The paper lacks a clear demonstration or ablation of FSMILES effectiveness, and the biological activity metric ECFP4 Tanimoto similarity (TS) is not clearly explained.
> >
> > ***response:*** We acknowledge the reviewer's desire for an ablation study on the effectiveness of the FSMILES representatio However, integrating the FSMILES representation, which was adopted from the widely cited lingo3dmol [a] and clearly referenced in our manuscript, required a substantial re-implementation of the underlying molecular representation backbone. Furthermore, our model relies on a large-scale GPT-based architecture trained on approximately 2 million molecules represented by FSMILES strings. Reverting this fundamental choice to perform a rigorous ablation (i.e., re-implementing the entire generation framework using a standard canonical SMILES representation and re-running all experiments) represents a significant development effort that is regrettably unfeasible within the tight timeframe of the rebuttal process and carries a considerable computational cost. We believe that the overall efficacy demonstrated by our model stands independently of a dedicated FSMILES ablation.
> >
> > Regarding the biological activity metric, the ECFP4 Tanimoto Similarity is a standard and widely accepted metric in chemoinformatics for quantifying the similarity of two molecules' potential biological activity. We specifically use this metric to evaluate whether our generated molecules possess the potential for biological activity. For the targets tested, which are drawn from the DUDE (Database of Useful Decoys: Enhanced) benchmark, there are numerous already-validated active molecules available. By calculating the ECFP4 Tanimoto Similarity between our generated molecules and these known active molecules for the corresponding target, we assess the likelihood that the generated compounds share the essential pharmacophoric features required for activity. A high ECFP4 TS value indicates that our generated molecule successfully recapitulates (or is similar to) a proven active scaffold. This metric serves as a crucial complement to purely physical metrics like docking scores, providing a necessary chemoinformatics-based validation that our generated molecules are not just physically stable in the pocket but are chemically plausible active compounds, thus offering a more holistic measure of the method's effectiveness.

---

> ### Author Response · Authors · 2025-11-22
>
> > Docking score: How is the initial binding pose determined when calculating min score? Can you provide the experiment affinity or docking scores for reference ligands?
>
> ***response:*** We have updated Tab. 1 in the revised manuscript to explicitly include the experimental binding affinity (or reported docking scores) for the reference ligands, which provides a direct benchmark for our results. Regarding the calculation of the "Min-in-place" score, the initial binding pose is the conformation predicted directly by our model. This pose is then subjected to local force field minimization (optimization) within the binding pocket to determine the final, optimized binding energy.
>
> > Sorting point clouds purely by coordinate order may disrupt spatial locality.
>
> ***response:*** Thank you for pointing it out. Actually, the use of sorting operations on point clouds is a standard practice in point-cloud generation and processing literature (e.g., Morton code-based ordering in PointGPT [b]). While sorting is heuristic, it provides a consistent and reproducible sequence for the model without substantially disrupting spatial relationships. Empirical evidence from prior works suggests that such sorting does not meaningfully harm the model’s ability to capture local or global geometric structure.
>
> > The paper does not explain how positional embeddings — a critical component of Transformer architectures — are implemented in this context.
>
> ***response:*** We thank the reviewer for raising this point. In our method, we adopt the default Transformer positional embeddings as used in GPT, which are learnable embeddings. We have clarified this detail in the revised manuscript to ensure transparency and reproducibility.
>
> > Table 1 does not report the average molecular weight of each model. ED2Mol tends to generate smaller molecules, which typically exhibit lower docking scores, lower strain energy, and higher drug-likeness. The paper only reports ED2Mol’s molecular weight, but ED2Mol performs weakly overall (Table 1). The authors should have compared against stronger baselines.
>
> ***response:*** We thank the reviewer for highlighting the importance of explicitly reporting the average molecular weight for all models.  We have reported QED, SAS, and Mol Weight as follows (ECloudGen$*$ denotes we use the ED extracted from ligand and conducted experiments for fair comparison, and Reference presents the results of experimentally validated bioactive ligands):
>
> Methods |QED |SAS | Molecule Weight
> |-|-|-|-|
> Pocket2Mol |0.56 |3.5 | 386
> TargetDiff |0.60 |4.0 |299
> Lingo3DMol |0.59 |3.1 | 348
> MolCRAFT | 0.51| 3.81|285
> ED2Mol |0.73 |3.9 |234
> ECloudGen |0.66 |2.9 |213
> ECloudGen $*$ |0.73 |2.9 |326
> EDMolGPT |0.57 |3.79 |385
> Reference |0.46 |3.6|438
>
> Specifically, while models like ED2Mol and ECloudGen achieve notably high average QED scores and favorable low SAS scores, they primarily generate significantly lower-molecular-weight compounds. This is substantially lower than both the Reference ligands and our EDMolGPT model.  In contrast, EDMolGPT maintains competitive drug-likeness metrics while generating molecules with an average molecular weight that is much closer to the complexity of the reference ligands, validating its capability to generate complex, relevant drug-like molecules.
>
> We have added the experiments in Sec.4.2

---

> > ### Author Response · Authors · 2025-11-22
> >
> > >Table 1 also omits important metrics such as diversity and novelty. Furthermore, Figure 6 only verifies that there is no data leakage between training and test sets, but does not confirm that the generated molecules are novel relative to the training data. Intuitively, for certain small molecules, using 199 points may provide excessive information about the reference ligand, effectively making the ligand-based design task easier. The authors need to demonstrate that their method can generate molecules that are significantly different from the reference ligand, rather than merely reproducing or slightly modifying it.
> >
> >
> > ***response:*** To quantitatively address the concern that our method might simply reproduce or slightly modify the reference ligand, we have calculated the structural diversity (Div) of the generated scaffolds relative to their respective reference ligands. As detailed in Table 2 (Ablation studies on temperature and resolution), the Div metric represents the average ECFP_TS Tanimoto Similarity score between the generated molecules and their corresponding reference ligands. Our results show consistently low Div scores (ranging from 0.176 to 0.186) across all tested conditions (different temperatures $T$ and resolution parameters $d_{\min}$).A Tanimoto similarity score in this range (around 0.18) indicates a very low structural resemblance between the generated molecules and the reference ligands. This robustly demonstrates that our model is not merely reproducing or slightly modifying the input ligand, but is successfully generating molecules with significantly different structural scaffolds, confirming the required structural novelty. Regarding the necessity to confirm novelty relative to the training data, we would like to clarify the distinction between the checks performed: Figure 6 confirms that there is no data leakage between the training set of protein-ligand pairs and the test set of target proteins, ensuring generation for unseen targets.
> >
> > Finally, the concern that using 199 points might provide "excessive information" is valid, but our use of a low-resolution, coarse-grained Electron Density (ED), as discussed in our previous response, specifically mitigates this risk. The low-resolution ED intentionally blurs the fine details of the reference ligand, transforming the 199 points from a rigid coordinate set into a soft spatial constraint. This constraint guides the generation toward the correct pocket space but leaves enough computational freedom for the model to generate the structurally diverse scaffolds demonstrated by the low Div scores in Table 2.
> >
> > [a] Feng, Wei, et al. "Generation of 3D molecules in pockets via a language model." Nature Machine Intelligence 6.1 (2024): 62-73.
> >
> > [b] Chen, Guangyan, et al. "Pointgpt: Auto-regressively generative pre-training from point clouds." Advances in Neural Information Processing Systems 36 (2023): 29667-29679.

---

> > > ### Comment · Reviewer_ApF4 · 2025-11-27
> > >
> > > Thanks for your detailed response. I'll consider raise my score after I carefully check your update.

---

### Official Review · Reviewer_mNj7 · 2025-10-31

**Soundness:** 3
**Presentation:** 3
**Contribution:** 2
**Rating:** 4
**Confidence:** 3

**Summary:**

The paper proposes EDMolGPT, a decoder-only (GPT-style) autoregressive model for structure-based drug design (SBDD) that generates 3D ligand molecules directly from low-resolution electron-density (ED) point clouds.

Unlike conventional SBDD approaches that condition on atom-level protein pocket structures or two-stage “ED→pocket→ligand” pipelines, EDMolGPT takes a 3D ED point cloud as the sole condition. Each sampled point is labeled with a pharmacophore type (e.g., H-bond donor/acceptor) to encode coarse chemical semantics. The point cloud and molecular tokens (fragmented SMILES and discretized 3D geometry tokens) are concatenated into one sequence, and a decoder-only Transformer is trained to predict the next molecular token autoregressively. During inference, the model generates ligand atoms and their relative coordinates step-by-step, constrained by predicted bond lengths, angles, and dihedrals.

On the DUD-E benchmark (101 targets), EDMolGPT achieves the highest bioactive recovery rate (41%) and competitive docking scores compared with Pocket2Mol, ED2Mol, Lingo3DMol, and MolCRAFT, while maintaining conformational stability without post-hoc relaxation.

Contributions:

- Introduces ED point clouds as a new conditional representation for SBDD that encodes both spatial and chemical context.

- Designs a decoder-only 3D molecular generator that unifies condition and generation in one autoregressive sequence model.

- Demonstrates strong performance and diversity on DUD-E, validating the feasibility of learning directly from physical electron-density signals.

**Strengths:**

Novel conditioning modality: Using electron density as the generative condition is a fresh, physically grounded idea that captures pocket flexibility and avoids hard atomic constraints.

Unified autoregressive formulation: The decoder-only design eliminates the need for a separate encoder, simplifying architecture and enabling efficient joint modeling of condition and ligand.

3D-aware tokenization: Integrating FSMILES with discretized coordinate and relative geometry tokens is elegant and practical for coupling chemical and spatial features.

Strong empirical results: Substantial gains on DUD-E demonstrate that electron-density conditioning provides meaningful guidance for 3D molecular generation.

**Weaknesses:**

Dependence on holo complexes: The conditioning ED maps are derived from known ligand–protein complexes, meaning the model is not applicable to apo pockets where no ligand is known. This limits its practical use in true de-novo design.

Pharmacophore labeling ambiguity: The pharmacophore features require ligand knowledge; the paper does not clarify how these labels could be inferred from ED alone during inference.

Arbitrary point-cloud ordering: Sorting ED points by xyz coordinates is heuristic and breaks rotational symmetry, which may hurt generalization.

Limited evaluation scope: Experiments are restricted to DUD-E; no validation on unseen protein families or cryo-EM-derived ED data.

Insufficient ablation and interpretability: The influence of pharmacophore labeling, sorting, and geometry tokens is under-analyzed; more visualization or ablation would strengthen claims.

**Questions:**

Inference scenario realism:
In practice, we often have only an apo pocket or predicted density map. How would EDMolGPT operate without a known holo-derived ED map? Could you approximate ED from the pocket atoms or use a learned ED predictor?

Pharmacophore derivation:
Since pharmacophore labels are computed from the known ligand during training, what is their source during inference? Are they predicted jointly, or assumed from pre-computed ED channels?

Permutation robustness:
How sensitive is the model to coordinate frame rotation or point-ordering perturbations? Would training with random permutations improve invariance?

Generalization to unseen systems:
Have you evaluated EDMolGPT on unseen protein families or on experimental cryo-EM densities to confirm cross-domain robustness?

Computational efficiency:
How does the autoregressive decoding speed and scaling behavior compare with diffusion-based SBDD models such as Pocket2Mol or TargetDiff?

---

> ### Author Response · Authors · 2025-11-22
>
> We sincerely appreciate your thoughtful feedback. In the following, your questions are answered point-by-point. We have updated the manuscript following your advice and highlighted in blue.
>
> > Dependence on holo complexes: The conditioning ED maps are derived from known ligand–protein complexes, meaning the model is not applicable to apo pockets where no ligand is known. This limits its practical use in true de-novo design.
>
> ***response:***  While the conditioning ED maps are indeed derived from known ligand–protein complexes, this does not fundamentally limit practical applicability. In structure-based drug discovery, conventional SBDD methods are also inherently restricted to holo pockets, since apo conformations without a bound ligand typically cannot provide reliable structural information due to substantial conformational changes upon ligand binding [a]. From this perspective, the applicability of our approach is comparable to standard SBDD workflows, which generate molecules based on available pocket representations.
>
> > Pharmacophore labeling ambiguity: The pharmacophore features require ligand knowledge; the paper does not clarify how these labels could be inferred from ED alone during inference.
>
> ***response:*** We understand the concern regarding the potential ambiguity of deriving pharmacophore labels from electron density alone. We would like to clarify that in our framework, the electron density is always generated from a known ligand conformation provided as input. As a result, the pharmacophore labels are directly computed from the ligand itself, rather than inferred from ED in an implicit or uncertain manner.
>
> In other words, the ED representation and the pharmacophore annotations originate from the same ligand source, ensuring consistency between the input and the labels at both training and inference. Therefore, the model does not need to deduce pharmacophore information solely from density patterns, and no additional inference step is required.
>
> > Arbitrary point-cloud ordering: Sorting ED points by xyz coordinates is heuristic and breaks rotational symmetry, which may hurt generalization.
>
> ***response:*** Thank you for pointing this out. We agree that point ordering is an important consideration for point-cloud representations. In our work, we follow a common practice adopted in recent point-cloud generation models (e.g., PointGPT [b] and related approaches), where the ED points are sorted using a spatial indexing heuristic such as Morton ordering. This ordering is not intended to encode geometric meaning, but rather to provide a stable and deterministic sequence that allows the transformer architecture to operate on point sets more effectively.
>
> > Limited evaluation scope: Experiments are restricted to DUD-E; no validation on unseen protein families or cryo-EM-derived ED data.
>
> ***response:*** We thank the reviewer for this insightful comment. While the main experiments focused on DUD-E, we additionally evaluated our method on experimentally measured cryo-EM-derived density maps. As shown in following, our approach is capable of generating valid molecules under these experimental conditions. Although the Min-in-place metric is slightly lower compared to the training data obtained via FFT-cutoff and inverse FFT, these results demonstrate that our method can generalize beyond the DUD-E dataset and remain effective on real density data, supporting its applicability to a broader range of protein targets.
>
> |Method|	Min-in-place|
> |-|-|
> cryo-EM-derived|	-5.90
> Ours|	-6.92
>
> We have added the additional experiments in Sec.4.2
>
> > Insufficient ablation and interpretability: The influence of pharmacophore labeling, sorting, and geometry tokens is under-analyzed; more visualization or ablation would strengthen claims.
>
> ***response:*** Thank you for pointing this out. We performed ablations on $N_p$ and on removing pharmacophore labels. As expected, increasing $N_p$ provides a more detailed description of the positive ED patterns, which improves the min-in-place score but slightly reduces diversity. Conversely, removing pharmacophore labels weakens geometric constraints, leading to higher diversity but a lower min-in-place score. The results are summarized below:
>
> | |Min-in-place |Div|
> |-|-|-|
> N_p=100 |-6.46|0.15|
> N_p=300 |-7.22|0.20|
> w/o pharmacophore labels|-6.15 |0.09|
>
> We have added the additional experiments in Sec.4.3
>
> Due to the tight review timeline and heavy computational cost of retraining, we were unable to further explore additional factors such as geometric tolerances, but we will include them in the camera-ready version if accepted. We also note that these ablations, while helpful, are not central to our main contribution: establishing the first ligand-based 3D structural generation paradigm driven by electron-density signals.

---

> > ### Author Response · Authors · 2025-11-22
> >
> > > Inference scenario realism: In practice, we often have only an apo pocket or predicted density map. How would EDMolGPT operate without a known holo-derived ED map? Could you approximate ED from the pocket atoms or use a learned ED predictor?
> >
> > ***response:*** We thank the reviewer for raising this point. In practice, our method focuses on holo pockets, where a bound ligand is known, and thus the conditioning ED maps are readily available—avoiding the scenario of relying solely on apo pockets or predicted densities. Approximating ED from pocket atoms or using a learned ED predictor could be possible in principle; however, due to the limited time during the rebuttal period, we have not explored this direction. Moreover, such approximations are likely to introduce accumulated errors that could significantly affect generation quality, and a careful study would be required to evaluate their feasibility.
> >
> > > Pharmacophore derivation: Since pharmacophore labels are computed from the known ligand during training, what is their source during inference? Are they predicted jointly, or assumed from pre-computed ED channels?
> >
> > ***response:*** Thank you for the insightful comment. During inference, we obtain pharmacophore labels in the same manner as in training—directly from the reference ligand’s electron density—rather than predicting them jointly. This is consistent with the holo setting, where a ligand is inherently present. Importantly, conventional SBDD methods also cannot operate on apo structures because the conformational differences between apo and holo pockets are typically too large for reliable pocket-based generation or docking; as a result, practical SBDD pipelines almost always assume access to a holo structure, which necessarily includes a ligand. Our method therefore, has the same applicability scope as SBDD: it works in realistic holo scenarios, while apo-based generation remains fundamentally unreliable.
> >
> > > Permutation robustness: How sensitive is the model to coordinate frame rotation or point-ordering perturbations? Would training with random permutations improve invariance?
> >
> > ***response:*** We thank the reviewer for raising this important point. In our method, we do not impose any specific bias when placing molecules in the coordinate frame. As a result, given sufficiently diverse and abundant training data, the model naturally learns to be insensitive to coordinate frame rotations and point-ordering permutations. To validate this, we performed a set of random rotations on input electron-density point clouds. As shown in following, the results indicate that the model’s performance remains stable under these rotations, suggesting that explicit training with random permutations is not strictly necessary in our current setting.
> > ||	Min-in-place|
> > |-|-|
> > rotation|	-6.88
> > Ours|	-6.92
> >
> >
> > > Generalization to unseen systems: Have you evaluated EDMolGPT on unseen protein families or on experimental cryo-EM densities to confirm cross-domain robustness?
> >
> > ***response:*** We thank the reviewer for this insightful comment. While the main experiments focused on DUD-E, we additionally evaluated our method on experimentally measured cryo-EM-derived density maps. As shown in follwing, our approach is capable of generating valid molecules under these experimental conditions. Although the Min-in-place metric is slightly lower compared to the training data obtained via FFT-cutoff and inverse FFT, these results demonstrate that our method can generalize beyond the DUD-E dataset and remain effective on real density data, supporting its applicability to a broader range of protein targets.
> >
> > |Method|	Min-in-place|
> > |-|-|
> > cryo-EM-derived|	-5.90
> > Ours|	-6.92
> >
> > > Computational efficiency: How does the autoregressive decoding speed and scaling behavior compare with diffusion-based SBDD models such as Pocket2Mol or TargetDiff?
> >
> > ***response:*** We thank the reviewer for the important query regarding computational efficiency. As shown in the Table, EDMolGPT utilizes an autoregressive GPT architecture conditioned by the electron density map, achieves a competitive average generation speed of approximately 1.5 seconds per molecule. For comparison, we have compiled the reported or estimated generation speeds for several prominent SBDD models. Pocket2Mol, another autoregressive model, reports a highly optimized generation speed of approximately 0.45 seconds per molecule. For diffusion-based models like TargetDiff, the speed depends heavily on the number of sampling steps (TargetDiff uses 1000 steps); the multi-step nature of the diffusion process typically makes them significantly slower than highly optimized autoregressive models.
> >
> > |Model|Architecture  |Avg. Generation Time (s/molecule) |
> > |-|-|-|
> > |ED-GPT|Autoregressive|$\approx1.5$|
> > |Pocket2Mol|Autoregressive|$\approx 0.45$|
> > |TargetDiff| Diffusion| $\approx 7$|
> >
> > We have added the additional experiments in Sec.E.3

---

> > > ### Author Response · Authors · 2025-11-22
> > >
> > > [a] Guo, Zuojun, et al. "Identification of protein–ligand binding sites by the level-set variational implicit-solvent approach." Journal of Chemical Theory and Computation 11.2 (2015): 753-765.
> > >
> > > [b] Chen, Guangyan, et al. "Pointgpt: Auto-regressively generative pre-training from point clouds." Advances in Neural Information Processing Systems 36 (2023): 29667-29679.

---

> > > > ### Comment · Reviewer_mNj7 · 2025-11-23
> > > > **Review Update**
> > > >
> > > > Thanks for the detailed response. I appreciate that the authors have released all the materials for reproducing this work. However,  I remain skeptical about this work's  contribution regarding its position compared to SBDD, particularly when considering the response `our method focuses on holo pockets, where a bound ligand is known, and thus the conditioning ED maps are readily available—avoiding the scenario of relying solely on apo pockets or predicted densities. Approximating ED from pocket atoms or using a learned ED predictor could be possible in principle`
> > > >
> > > > Thus, I decide to keep the score unchanged.

---

> > > > > ### Author Response · Authors · 2025-11-23
> > > > >
> > > > > Thanks for the response. We would like to further clarify our contribution and its applicability relative to SBDD.
> > > > >
> > > > > First, regarding the reviewer’s concern about relying on holo pockets: **in practical drug discovery pipelines, existing SBDD approaches also predominantly operate on holo structures, because the structural discrepancy between apo and holo pockets is often large and unpredictable[a]. In fact, no real medicinal chemistry workflow would start drug design purely from an apo structure.**  Therefore, conditioning on a holo conformation is consistent with both realistic practice and standard SBDD settings. **From this perspective, the applicability scope of our method is fully aligned with existing SBDD methodologies.**
> > > > >
> > > > > Second, our work takes an important step forward in ligand-based structure generation by explicitly leveraging electron density as conditioning information. **To the best of our knowledge, no prior work has performed controllable 3D molecular generation under ligand structure constraints, leading to only a limited number of relevant baselines.** Although our method is not a conventional SBDD technique, we still perform fair comparisons with representative methods. Notably, Table 1 shows that when ECloudGen is evaluated on ligand-derived ED, our model still achieves significantly better performance.
> > > > >
> > > > >  We hope this clarification helps better situate our contribution.
> > > > >
> > > > > [a] Guo, Zuojun, et al. "Identification of protein–ligand binding sites by the level-set variational implicit-solvent approach." Journal of Chemical Theory and Computation 11.2 (2015): 753-765.

---

### Official Review · Reviewer_gefv · 2025-10-31

**Soundness:** 2
**Presentation:** 2
**Contribution:** 2
**Rating:** 4
**Confidence:** 3

**Summary:**

The paper conditions generation on the low-resolution electron-density point cloud of a known ligand and employs a decoder-only autoregressive framework to produce 3D molecules. While results on DUD-E show some advantages, reproducibility and evaluation fairness are problematic.

**Strengths:**

Using a low-resolution electron-density point cloud as the conditioning signal and a pure decoder autoregressive architecture to directly generate molecules with 3D conformations is a simple, efficient idea that is engineering-friendly and scalable.

**Weaknesses:**

1) Reproducibility is insufficient: no released code, model weights, data processing or evaluation scripts, and missing environment and random seed settings, and heavy reliance on commercial software makes the results difficult to reproduce.
2) The training set’s exact sources, versions, licenses, and cleaning/normalization procedures are not specified.
The method conditions on the reference ligand’s ED, which is essentially ligand-based generation, yet it is directly compared against SBDD baselines conditioned on protein pockets; this is not a fair comparison.
3) The ED is derived from the ligand rather than the protein/complex, making it difficult to substantiate claims about capturing pocket flexibility or avoiding rigid pocket assumptions; this reflects a conceptual mismatch.
4) The method conditions on the reference ligand’s ED, which is essentially ligand-based generation, yet it is directly compared against SBDD baselines conditioned on protein pockets, this is not a fair comparison.
5) Ablations are incomplete: there is no systematic study of the number of sampled points Np, coordinate quantization step σ, tolerances for relative geometry , or the effect of turning pharmacophore labels on/off.

**Questions:**

Code and model availability: You currently do not provide training/inference code, pretrained weights, evaluation scripts, or environment configuration, making it impossible to reproduce key results. Suggestion: release a complete pipeline in an anonymous repository (training, inference, evaluation), including Docker/conda environments, random seed and determinism settings, logs and hyperparameters; provide pretrained weights and a small set of example data.
Training data provenance and cleaning: The statement “~8M → ~2M after filtering” lacks detail on specific sources (e.g., ChEMBL/ZINC/PubChem), versions, download dates, licenses, deduplication, and standardization (salt stripping, stereochemistry/tautomer handling, normalization). Please document these aspects and provide the corresponding scripts and statistics.

---

> ### Author Response · Authors · 2025-11-22
>
> We sincerely appreciate your thoughtful feedback. In the following, your questions are responsed point-by-point. We have updated the manuscript following your advice and highlighted in blue.
>
> > Reproducibility is insufficient: no released code, model weights, data processing or evaluation scripts, and missing environment and random seed settings, and heavy reliance on commercial software makes the results difficult to reproduce.
>
> ***response:*** Thank you for pointing this out. At the current stage, due to internal approval requirements, we are only able to release the testing code at our anonymous link: \url{https://anonymous.4open.science/r/EDMolGPT/}
> . We had planned to provide model weights as well, but the model size exceeds the upload limit of the anonymous repository.
>
> We sincerely commit that upon acceptance, we will release:
>
> (1) the complete training and inference code,
>
> (2) model weights,
>
> (3) full environment configuration files,
>
> (4) random seeds,
>
> (5) and all data processing and evaluation scripts.
>
> Regarding the use of commercial software (Glide), we note that this is common practice in prior work—for example, Lingo3Dmol [a] also reports results based on Glide. Glide typically provides more reliable scoring than open-source alternatives such as Vina, and thus allows a more rigorous evaluation of our method.
>
> > The training set’s exact sources, versions, licenses, and cleaning/normalization procedures are not specified. The method conditions on the reference ligand’s ED, which is essentially ligand-based generation, yet it is directly compared against SBDD baselines conditioned on protein pockets; this is not a fair comparison.
>
> ***response:*** Thank you for the valuable comments. The training data we used is publicly available at \url{http://data.aicnic.cn/dms-html/dataset_detail.html?id=848}
> , and by following the full workflow provided on that page, one can obtain the complete training set. Upon acceptance, we will release our full preprocessing pipeline, including cleaning, normalization, and data extraction. In the current anonymous repository \url{https://anonymous.4open.science/r/EDMolGPT/}, we have already included several example cases (point clouds and corresponding ligands extracted from the DUD-E dataset) to provide a concrete reference for our data handling.
>
> We fully understand the reviewer’s concern regarding the comparison setup. To the best of our knowledge, our work is the first to perform ligand-based electron-density (ED)–conditioned structure generation, and thus there is no existing ligand-based baseline for a strictly fair comparison. We therefore compare against representative SBDD methods conditioned on protein pockets, not because the settings are identical, but because these are the only available approaches whose outputs and evaluation metrics can be meaningfully contrasted with ours. **In addition, we also report the results of ECloudGen with the electron density extracted from the corresponding ligand in Tab.1 in the paper, and the results also verify the effectiveness of our method.**
>
> We hope that our work can serve as an initial step toward building this new ligand-based generation paradigm and provide a foundation and reference for future research in this direction.

---

> > ### Author Response · Authors · 2025-11-22
> >
> > > The ED is derived from the ligand rather than the protein/complex, making it difficult to substantiate claims about capturing pocket flexibility or avoiding rigid pocket assumptions; this reflects a conceptual mismatch.
> >
> > ***response:*** We appreciate the reviewer's insightful comment. The experimentally determined conformation of the ligand is inherently and significantly influenced by the spatial constraints and key transient interactions within the binding pocket. By using the ligand's electron density as the defining feature, we are effectively utilizing the ligand's final, constrained state as a high-fidelity proxy for the critical spatial and energetic boundaries of the binding site.
> >
> > A key aspect of our methodology is the intentional use of a low-resolution, coarse-grained representation of the electron density. This coarse-graining process smooths out the fine details present in high-resolution data, which crucially prevents the model from being rigidly locked into a single, hyper-precise conformation defined by a single static pocket structure. By offering a "softer," less defined constraint, the low-resolution ED provides the simulated system with sufficient degrees of freedom to explore alternative, energetically viable conformational states around the experimentally observed density. This method allows the model to computationally sample the transient flexibility of the binding site within the boundaries dictated by the ligand's experimentally constrained position. In summary, while the ED is derived from the ligand, we assert that the ligand's constrained conformation serves as a valid proxy for pocket geometry, and the intentional use of a low-resolution, coarse-grained ED successfully introduces the necessary computational freedom to avoid a rigid pocket assumption, aligning conceptually with our goal of capturing flexibility.
> >
> > **We also provide a detailed example to further validate our claim. As shown in Fig.1 in the paper, biochemical assays confirm the activity of ligands bearing bulky substituents at this site, indicating that the binding site is conformationally flexible and not comprehensively represented by the static structure. Accordingly, the generation constrained by the static pocket fails to produce these active compounds. In contrast, our low-resolution ED–guided approach accommodates local conformational plasticity, enabling the successful generation of bulky, yet active, substituents.**
> >
> > >The method conditions on the reference ligand’s ED, which is essentially ligand-based generation, yet it is directly compared against SBDD baselines conditioned on protein pockets, this is not a fair comparison.
> >
> > ***response:*** We thank the reviewer for this insightful comment. Indeed, our method operates solely from the reference ligand’s electron density, making it fundamentally a ligand-based generation approach, whereas structure-based drug design (SBDD) baselines have full access to the protein environment. Consequently, when evaluating pocket-dependent metrics, a direct comparison is inherently unfair, as the conditions differ between the methods.
> >
> > However, for metrics that do not rely on pocket information—such as the recovery of active molecules—our evaluation is fully fair and meaningful. Importantly, to the best of our knowledge, our work is the first to perform fully structure-aware ligand-based generation. There are currently no prior methods that generate new drug-like molecules directly from a ligand structure alone. Therefore, despite the differences in conditioning, comparison with existing SBDD methods remains the only viable option to contextualize the performance of our approach. **In addition, we also report the results of ECloudGen with the electron density extracted from the corresponding ligand, and the results also verify the effectiveness of our method.**
> >
> > Moreover, as discussed earlier, ligand-based structure design is fundamentally important, and we believe that our work can provide foundational support and serve as a reliable baseline for future drug discovery and design efforts.

---

> > > ### Author Response · Authors · 2025-11-22
> > >
> > > >Ablations are incomplete: there is no systematic study of the number of sampled points Np, coordinate quantization step σ, tolerances for relative geometry , or the effect of turning pharmacophore labels on/off.
> > >
> > > ***response:*** Thank you for pointing this out. We performed ablations on $N_p$ and on removing pharmacophore labels. As expected, increasing $N_p$ provides a more detailed description of the positive ED patterns, which improves the min-in-place score but slightly reduces diversity. Conversely, removing pharmacophore labels weakens geometric constraints, leading to higher diversity but a lower min-in-place score. The results are summarized below:
> > >
> > > | |Min-in-place |Div|
> > > |-|-|-|
> > > N_p=100 |-6.46|0.15|
> > > N_p=300 |-7.22|0.20|
> > > w/o pharmacophore labels|-6.15 |0.09|
> > >
> > >
> > > Due to the tight review timeline and heavy computational cost of retraining, we were unable to further explore additional factors such as coordinate quantization or geometric tolerances, but we will include them in the camera-ready version. We also note that these ablations, while helpful, are not central to our main contribution: establishing the first ligand-based 3D structural generation paradigm driven by electron-density signals.
> > >
> > > We have added the additional experiments in Sec.4.3.
> > >
> > > > Code and model availability: You currently do not provide training/inference code, pretrained weights, evaluation scripts, or environment configuration, making it impossible to reproduce key results. Suggestion: release a complete pipeline in an anonymous repository (training, inference, evaluation), including Docker/conda environments, random seed and determinism settings, logs and hyperparameters; provide pretrained weights and a small set of example data. Training data provenance and cleaning: The statement “~8M → ~2M after filtering” lacks detail on specific sources (e.g., ChEMBL/ZINC/PubChem), versions, download dates, licenses, deduplication, and standardization (salt stripping, stereochemistry/tautomer handling, normalization). Please document these aspects and provide the corresponding scripts and statistics.
> > >
> > > ***response:*** We appreciate the reviewer’s thoughtful suggestions regarding reproducibility and data transparency. Due to institutional approval constraints at this stage, we are only able to release partial testing code in the anonymous repository. Upon acceptance, we will make the entire pipeline publicly available, including full training and inference code, evaluation scripts, pretrained weights, complete environment configuration files (Docker/conda), random seed settings, logs, and all hyperparameters,  to ensure fully deterministic reproducibility. Regarding data provenance, the training data used in our work is obtained from the publicly accessible resource at \url{http://data.aicnic.cn/dms-html/dataset_detail.html?id=848}; following the workflow on this page yields the complete dataset. The current version of the anonymous repository already contains several example cases to illustrate our data extraction format. For the full dataset (from ~8M raw entries to ~2M after filtering), we will release detailed documentation describing data sources, versions, download dates, licenses, deduplication rules, salt/ion removal, stereochemistry and tautomer normalization procedures, as well as all statistical summaries and processing scripts. We will ensure that every step of filtering and standardization is fully transparent and reproducible in the final code release.
> > >
> > > [a] Feng, Wei, et al. "Generation of 3D molecules in pockets via a language model." Nature Machine Intelligence 6.1 (2024): 62-73.

---

### Author Response · Authors · 2025-11-24
**Clarification regarding comparison with SBDD**

Dear Reviewers,

We sincerely thank all reviewers for their constructive comments and valuable discussion. Here we would like to clarify the positioning and applicability of our work, particularly in relation to existing structure-based drug design (SBDD) paradigms.

First, from a practical, real-world standpoint, modern drug discovery workflows operate on holo pockets rather than apo pockets. ***The structural difference between apo and holo conformations is often large and unpredictable, so both academic and industrial SBDD practices almost always rely on holo (ligand-bound) structures when available.*** In real medicinal-chemistry practice, researchers faced with a new target typically first identify a functional pocket by obtaining a binder in it. This step helps the research team to confirm that this is really a pocket and, at the same time, determine the holo conformation. This bound complex might be derived from a native binder or a tool compound obtained through bioactivity-based screening. Then, the novel molecule design starts to focus on obtaining a more potent scaffold with robust pharmacological properties. ***Thus, in terms of intended application scope and practical usage, our method and conventional SBDD approaches are consistent.***

To make the relationship clearer: ***many SBDD methods take a holo complex and remove the ligand to obtain a target pocket for design (i.e., they keep the pocket and generate or optimize ligands to fill it). By contrast, our ligand-based generation flips this conditioning: we remove the pocket and use the ligand (or ligand-derived electron density) as the conditioning signal to generate structures.*** Both strategies operate in the holo setting and address the same practical design scenario, but they condition on complementary parts of the same holo complex.

We also acknowledge the reviewers’ concern about the scarcity of directly comparable baselines. Because this is, to our knowledge, ***the first work to perform controllable ligand-based 3D generation under electron-density constraints, there are very few prior methods that target the same problem formulation.*** For fair comparison, we therefore evaluated representative SBDD-style and density-based models where possible. In particular, Table 1 reports experiments where ECloudGen was evaluated using ligand-derived ED — and our method shows substantial improvements under these settings. We believe these comparisons, while constrained by the limited baseline set, demonstrate the effectiveness of incorporating ED-conditioned generation.

Finally, we have carefully addressed reviewers’ suggestions and ***revised the manuscript accordingly***, adding clarifying details, additional experiments, and discussion where requested.

Best regards,

Author 4236

---

### Meta-Review · Area_Chair_jAmm · 2025-12-09

**Summary:**

This paper proposes EDMolGPT, a decoder-only framework that utilizes electron density (ED) as a conditional signal to generate 3D drug molecules efficiently.

### Pros
* Using electron density as a generative condition is an innovative approach
* The proposed method is efficient
* The presentation is good and easy-to-follow

### Cons

* Unfair comparisons
* Limited applicability
* Reproducibility is not good
* missing baselines

### AC's evaluation

1. from reviews and rebuttals

This paper receives 442, all reviewers vote for rejection.

2. from AC's reading

While the engineering approach (Decoder-only + ED) is interesting, the paper suffers from a conceptual mismatch. It frames itself as an SBDD solution but utilizes strong ligand priors (electron density), effectively turning the task into shape-based matching. Comparing this approach directly to prior-free SBDD baselines like TargetDiff is methodologically unsound. Furthermore, the dependency on holo structures limits the tool to lead optimization scenarios, contradicting its broader de novo claims. The authors should reposition the work  and compare against appropriate baselines in a future submission.

**Reviewer Concerns:**

outstanding concerns:

* Unfair comparison (Critical - Reviewer gefv): This is the fatal flaw. The paper compares EDMolGPT (which implicitly uses reference ligand information via ED) against pure SBDD methods like Pocket2Mol and TargetDiff (which only use protein pocket information). Reviewers argue this is an "apples-to-oranges" comparison, as the former task is significantly easier due to the leakage of ligand priors.

* Limited applicability (Reviewer mNj7): The method relies heavily on holo crystal structures to extract electron density. This restricts its utility in de novo design for apo pockets (targets without known binders), limiting its real-world impact.

* Reproducibility issues (Reviewer gefv): The data preprocessing pipeline is poorly documented, and reliance on proprietary software (Glide) hinders reproducibility.

* Missing baselines (Reviewer ApF4): The paper lacks comparison with other electron-density-based generation methods (e.g., ED2Mol), making it difficult to assess the specific contribution of the proposed architecture.

**Reviewer Scores:**

I do not think anyone will increase scores.

---

### Decision · Program_Chairs · 2026-01-26

Reject